Proceedings of the 7th Symposium on Advances in Approximate Bayesian Inference, 2025 1–40

# Normalizing Flow Regression for Bayesian Inference with Offline Likelihood Evaluations

**Chengkun Li**[1]                                    CHENGKUN.LI@HELSINKI.FI
**Bobby Huggins**[2]                                      B.HUGGINS@WUSTL.EDU
**Petrus Mikkola**[1]                                  PETRUS.MIKKOLA@HELSINKI.FI
**Luigi Acerbi**[1]                                     LUIGI.ACERBI@HELSINKI.FI

[1]*Department of Computer Science, University of Helsinki*

[2]*Department of Computer Science and Engineering, Washington University in St. Louis*

## Abstract

Bayesian inference with computationally expensive likelihood evaluations remains a significant challenge in many scientific domains. We propose normalizing flow regression (NFR), a novel offline inference method for approximating posterior distributions. Unlike traditional surrogate approaches that require additional sampling or inference steps, NFR directly yields a tractable posterior approximation through regression on existing log-density evaluations. We introduce training techniques specifically for flow regression, such as tailored priors and likelihood functions, to achieve robust posterior and model evidence estimation. We demonstrate NFR's effectiveness on synthetic benchmarks and real-world applications from neuroscience and biology, showing superior or comparable performance to existing methods. NFR represents a promising approach for Bayesian inference when standard methods are computationally prohibitive or existing model evaluations can be recycled.

## 1. Introduction

Black-box models of varying complexity are widely used in scientific and engineering disciplines for tasks such as parameter estimation, hypothesis testing, and predictive modeling (Sacks et al., 1989; Kennedy and O'Hagan, 2001). Bayesian inference provides a principled framework for quantifying uncertainty in both parameters and models by computing full posterior distributions and model evidence (Gelman et al., 2013). However, Bayesian inference is often analytically intractable, requiring the use of approximate methods like Markov chain Monte Carlo (MCMC; Brooks, 2011) or variational inference (VI; Blei et al., 2017). These methods typically necessitate repeated evaluations of the target density, and many require differentiability of the model (Neal, 2011; Kucukelbir et al., 2017). When model evaluations are computationally expensive – for instance, involving extensive numerical methods – these requirements make standard Bayesian approaches impractical.

Due to these computational demands, practitioners often resort to simpler alternatives such as maximum a posteriori (MAP) estimation or maximum likelihood estimation (MLE);[1] see for example Wilson and Collins (2019); Ma et al. (2023). While these point estimates can provide useful insights, they fail to capture parameter uncertainty, potentially leading to overconfident or biased conclusions (Gelman et al., 2013). This limitation highlights the need for efficient posterior approximation methods that avoid the computational costs of standard inference techniques.

---

1. In practice, MLE corresponds to MAP with flat priors.

Recent advances in surrogate modeling present promising alternatives for addressing these challenges. Costly likelihood or posterior density functions are efficiently approximated via *surrogates* such as Gaussian processes (GPs; Rasmussen, 2003; Gunter et al., 2014; Acerbi, 2018, 2019; Järvenpää et al., 2021; Adachi et al., 2022; El Gammal et al., 2023). To mitigate the cost of standard GPs, both sparse GPs and deep neural networks have also served as surrogates for posterior approximation (Wang et al., 2022; Li et al., 2024). However, these approaches share a key limitation: the obtained surrogate model, usually of the log likelihood or log posterior, does not directly provide a valid probability distribution. Additional steps, such as performing MCMC or variational inference on the surrogate, are needed to yield tractable posterior approximations. Furthermore, many of these methods require active collections of new likelihood evaluations, which might be unfeasible or wasteful of existing evaluations.

To address these challenges, we propose using *normalizing flows* as regression models for directly approximating the posterior distribution from *offline* likelihood or density evaluations. While normalizing flows have been extensively studied for variational inference (Rezende and Mohamed, 2015; Agrawal et al., 2020), density estimation (Dinh et al., 2017), and simulation-based inference (Lueckmann et al., 2021; Radev et al., 2022), their application as *regression models* for posterior approximation remains largely unexplored. Unlike other surrogate methods, normalizing flows directly yield a tractable posterior distribution which is easy to evaluate and sample from. Moreover, unlike other applications of normalizing flows, our regression approach is *offline*, recycling existing log-density evaluations (e.g., from MAP optimizations as in Li et al., 2024) rather than requiring costly new evaluations from the target model.

The main contribution of this work consists of proposing normalizing flows as a regression model for surrogate-based, offline Bayesian inference, together with techniques for training them in this context, such as sensible priors over flows. We demonstrate the effectiveness of our method on challenging synthetic and real-world problems, showing that normalizing flows can accurately estimate posterior distributions and their normalizing constants through regression. This work contributes a new approach for Bayesian inference in settings where standard methods are computationally prohibitive, affording more robust and uncertainty-aware modeling across scientific and engineering applications.

## 2. Background

### 2.1. Normalizing flows

Normalizing flows construct flexible probability distributions by iteratively transforming a simple *base distribution*, typically a multivariate Gaussian distribution. A normalizing flow defines an invertible transformation $T_\phi : \mathbb{R}^D \to \mathbb{R}^D$ with parameters $\phi$. Let $\mathbf{u} \in \mathbb{R}^D$ be a random variable from the base distribution $p_{\mathbf{u}}$. For a random variable $\mathbf{x} = T_\phi(\mathbf{u})$, the change of variables formula gives its density as:

$$q_\phi(\mathbf{x}) = p_{\mathbf{u}}(\mathbf{u}) \left| \det J_{T_\phi}(\mathbf{u}) \right|^{-1}, \quad \mathbf{u} = T_\phi^{-1}(\mathbf{x}), \tag{1}$$

where $J_{T_\phi}$ denotes the Jacobian matrix of the transformation. The transformation $T_\phi(\mathbf{u})$ can be designed to balance expressive power with efficient computation of its Jacobian determinant. In this paper, we use the popular masked autoregressive flow (MAF; Papamakarios

et al., 2017). MAF constructs the transformation through an autoregressive process, where each component $\mathbf{x}^{(i)}$ depends on previous components through:

$$\mathbf{x}^{(i)} = g_{\text{scale}}(\alpha^{(i)}) \cdot \mathbf{u}^{(i)} + g_{\text{shift}}(\mu^{(i)}). \tag{2}$$

Here, $g_{\text{scale}}$ is typically chosen as the exponential function to ensure positive scaling, while $g_{\text{shift}}$ is usually the identity function. The parameters $\alpha^{(i)}$ and $\mu^{(i)}$ are outputs of unconstrained scalar functions $h_\alpha$ and $h_\mu$ that take the preceding components as inputs:

$$\alpha^{(i)} = h_\alpha(\mathbf{x}^{(1:i-1)}), \quad \mu^{(i)} = h_\mu(\mathbf{x}^{(1:i-1)}), \tag{3}$$

where $h_\alpha$ and $h_\mu$ are usually parametrized by neural networks with parameters $\boldsymbol{\phi}$.

This autoregressive structure ensures invertibility of the transformation and enables efficient computation of the Jacobian determinant needed for the density calculation in Eq. 1 (Papamakarios et al., 2021). To accelerate computation, MAF is implemented in parallel via masking, using a neural network architecture called Masked AutoEncoder for Distribution Estimation (MADE; Germain et al. 2015).

## 2.2. Bayesian inference

Bayesian inference provides a principled framework for inferring unknown parameters $\mathbf{x}$ given observed data $\mathcal{D}$. From Bayes' theorem, the posterior distribution $p(\mathbf{x}|\mathcal{D})$ is:

$$p(\mathbf{x}|\mathcal{D}) = \frac{p(\mathcal{D}|\mathbf{x})p(\mathbf{x})}{p(\mathcal{D})}, \tag{4}$$

where $p(\mathcal{D}|\mathbf{x})$ is the likelihood, $p(\mathbf{x})$ is the prior over the parameters, and $p(\mathcal{D})$ is the normalizing constant, also known as evidence or marginal likelihood, a quantity useful in Bayesian model selection (MacKay, 2003). Two widely used approaches for approximating this posterior are variational inference and Markov chain Monte Carlo (Gelman et al., 2013).

VI turns posterior approximation into an optimization problem by positing a family of parametrized distributions, such as normalizing flows ($q_{\boldsymbol{\phi}}$ in Section 2.1), and optimizing over the parameters $\boldsymbol{\phi}$. The objective to maximize is commonly the evidence lower bound (ELBO), which is equivalent to minimizing the Kullback-Leibler (KL) divergence between the approximate distribution and $p(\mathbf{x}|\mathcal{D})$(Blei et al., 2017). When the likelihood $p(\mathcal{D}|\mathbf{x})$ is a black box, the estimated ELBO gradients can exhibit high variance, thus requiring many evaluations to converge (Ranganath et al., 2014). MCMC methods, such as Metropolis-Hastings, aim to draw samples from the posterior by constructing a Markov chain that converges to $p(\mathbf{x}|\mathcal{D})$. While MCMC offers asymptotic guarantees, it requires many likelihood evaluations. Due to the typically large number of required evaluations, both VI and MCMC are often infeasible for black-box models with expensive likelihoods.

## 3. Normalizing Flow Regression

We now present our proposed method, Normalizing Flow Regression (NFR) for approximate Bayesian posterior inference. In the following, we denote with $\mathbf{X} = (\mathbf{x}_1, \ldots, \mathbf{x}_N)$ a set of input locations where we have evaluated the target posterior, with corresponding unnormalized log-density evaluations $\mathbf{y} = (y_1, \ldots, y_N)$, where $\mathbf{x}_n \in \mathbb{R}^D$ and $y_n \in \mathbb{R}$. Evaluations

have associated observation noise $\boldsymbol{\sigma}^2 = (\sigma_1^2, \ldots, \sigma_N^2),$[2] where we set $\sigma_n^2 = \sigma_{\min}^2 = 10^{-3}$ for noiseless cases. We collect these into a training dataset $\boldsymbol{\Xi} = (\mathbf{X}, \mathbf{y}, \boldsymbol{\sigma}^2)$ for our flow regression model. Throughout this section, we use $p_{\text{target}}(\mathbf{x}) \equiv p(\mathcal{D}|\mathbf{x})p(\mathbf{x})$ to denote the *unnormalized* target posterior density.

## 3.1. Overview of the regression model

We use a normalizing flow $T_{\boldsymbol{\phi}}$ with normalized density $q_{\boldsymbol{\phi}}(\mathbf{x})$ to fit $N$ observations of the log density of an unnormalized target $p_{\text{target}}(\mathbf{x})$, using the dataset $\boldsymbol{\Xi} = (\mathbf{X}, \mathbf{y}, \boldsymbol{\sigma}^2)$. Let $f_{\boldsymbol{\phi}}(\mathbf{x}) = \log q_{\boldsymbol{\phi}}(\mathbf{x})$ be the flow's log-density at $\mathbf{x}$. The log-density *prediction* of our regression model is:

$$f_{\boldsymbol{\psi}}(\mathbf{x}) = f_{\boldsymbol{\phi}}(\mathbf{x}) + C, \tag{5}$$

where $C$ is an additional free parameter accounting for the unknown (log) normalizing constant of the target posterior. The parameter set of the regression model is $\boldsymbol{\psi} = (\boldsymbol{\phi}, C)$.

We train the flow regression model itself via MAP estimation, by maximizing:

$$\begin{aligned}\mathcal{L}(\boldsymbol{\psi}) = &\ \log p(\mathbf{y}|\mathbf{X}, \boldsymbol{\sigma}^2, f_{\boldsymbol{\phi}}, C) + \log p(\boldsymbol{\phi}) + \log p(C) \\ = &\ \sum_{n=1}^N \log p\left(y_n | f_{\boldsymbol{\psi}}(\mathbf{x}_n), \sigma_n^2\right) + \log p(\boldsymbol{\phi}) + \log p(C),\end{aligned} \tag{6}$$

where $p\left(y_n | f_{\boldsymbol{\psi}}(\mathbf{x}_n), \sigma_n^2\right)$ is the likelihood of observing log-density value $y_n,$[3] while $p(\boldsymbol{\phi})$ and $p(C)$ are priors over the flow parameters and log normalizing constant, respectively.

Since we only have access to finite pointwise evaluations of the target log-density, $\boldsymbol{\Xi} = (\mathbf{X}, \mathbf{y}, \boldsymbol{\sigma}^2)$, the choice of the likelihood function and priors for the regression model is crucial for accurate posterior approximation. We detail these choices in Sections 3.2 and 3.3.

## 3.2. Likelihood function for log-density observations

For each observation $y_n$, let $f_n \equiv p_{\text{target}}(\mathbf{x}_n)$ denote the true unnormalized log-density value, which our flow regression model aims to estimate via its prediction $f_{\boldsymbol{\psi}}(\mathbf{x}_n)$. We now discuss how to choose an appropriate likelihood function $p\left(y_n \mid f_{\boldsymbol{\psi}}(\mathbf{x}_n), \sigma_n^2\right)$ for these log-density observations. A natural first choice would be a Gaussian likelihood,

$$p(y_n | f_{\boldsymbol{\psi}}(\mathbf{x}_n), \sigma_n^2) = \mathcal{N}(y_n | f_{\boldsymbol{\psi}}(\mathbf{x}_n), \sigma_n^2). \tag{7}$$

However, this choice has a significant drawback emerging from the fact that maximizing this likelihood corresponds to minimizing the point-wise squared error $|y_n - f_{\boldsymbol{\psi}}(\mathbf{x}_n)|^2/\sigma_n^2$. Since log-density values approach negative infinity as density values approach zero, small errors in near-zero density regions of the target posterior would dominate the regression objective in Eq. 6. This would cause the normalizing flow to overemphasize matching these

---

2. Log-density observations can be noisy when likelihood calculation involves simulation or Monte Carlo methods. Noise for each observation can then be quantified independently via bootstrap or using specific estimators (van Opheusden et al., 2020; Acerbi, 2020; Järvenpää et al., 2021).

3. Assuming conditionally independent noise on the log-density estimates, which holds trivially for noiseless observations and for many estimation methods (van Opheusden et al., 2020; Järvenpää et al., 2021).

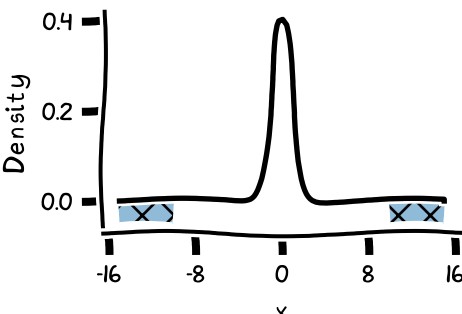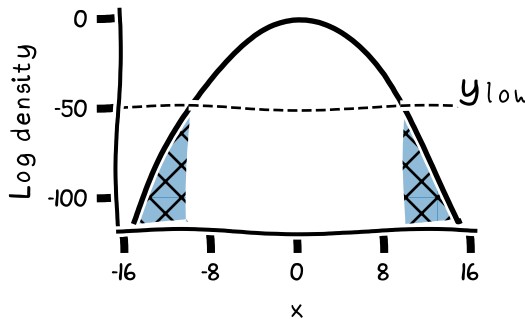

Figure 1: Illustration of the censoring effect of the Tobit likelihood on a target density. The left panel shows the density plot, while the right panel displays the corresponding log-density values. The shaded region represents the censored observations with log-density values below $y_{\text{low}}$, where the density is near-zero.

near-zero density observations at the expense of accurately modeling the more important high-density regions.

To address this issue, we propose a more robust *Tobit likelihood* for flow regression, inspired by the Tobit model (Amemiya, 1984) and the *noise shaping* technique (Li et al., 2024). Let $f_{\max} \equiv \max_{\mathbf{x}} \log p(\mathbf{x})$ denote the maximum log-density value (i.e., at the distribution mode). The Tobit likelihood takes the form:

$$p(y_n|f_{\boldsymbol{\psi}}(\mathbf{x}_n), \sigma_n^2) = \begin{cases} \mathcal{N}\left(y_n; f_{\boldsymbol{\psi}}(\mathbf{x}_n), \sigma_n^2 + s(f_{\max} - f_n)^2\right) & \text{if } y_n > y_{\text{low}}, \\ \Phi\left(\frac{y_{\text{low}} - f_{\boldsymbol{\psi}}(\mathbf{x}_n)}{\sqrt{\sigma_n^2 + s(f_{\max} - f_n)^2}}\right) & \text{if } y_n \leq y_{\text{low}}, \end{cases} \tag{8}$$

where $y_{\text{low}}$ represents a threshold below which we censor observed log-density values, $\Phi$ is the standard normal cumulative distribution function (CDF), and $s(\cdot)$ a noise shaping function, discussed below. When $y_n \leq y_{\text{low}}$, the Tobit likelihood only requires the model's prediction $f_{\boldsymbol{\psi}}(\mathbf{x}_n)$ to fall below $y_{\text{low}}$, rather than match $y_n$ exactly (see Figure 1). The function $s(\cdot)$ acts as a noise shaping mechanism (Li et al., 2024) that linearly increases observation uncertainty for lower-density regions, further retaining information from low-density observations without overfitting to them (see Appendix A.1 for details).

### 3.3. Prior settings

The flow regression model's log-density prediction depends on both the flow parameters $\boldsymbol{\phi}$ and the log normalizing constant $C$ (Eq. 5), leading to a non-identifiability issue. Given a sufficiently expressive flow, alternative parameterizations $(\boldsymbol{\phi}', C')$ can yield identical predictions at observed points. While this suggests the necessity of informative priors for both the flow and the normalizing constant, setting a meaningful prior on $C$ is challenging since the target density evaluations are neither i.i.d. nor samples from the target distribution. Therefore, we focus on imposing sensible priors on the flow parameters $\boldsymbol{\phi}$, which indirectly regularize the normalization constant and avoid the pitfalls of complete non-identifiability.

A normalizing flow consists of a base distribution and transformation layers. The base distribution can incorporate prior knowledge about the target posterior's shape, for instance

from a moment-matching approximation. In our case, the training data $\boldsymbol{\Xi} = (\mathbf{X}, \mathbf{y}, \boldsymbol{\sigma}^2)$ comes from MAP optimization runs on the target posterior. We use a multivariate Gaussian with diagonal covariance as the base distribution $p_0$, and estimate its mean and variance along each dimension using the sample mean and variance of observations with sufficiently high log-density values (see Appendix A.1 for further details).

Specifying priors for the flow transformation layers is less straightforward since they are parameterized by neural networks (Fortuin, 2022). As a normalizing flow is itself a distribution, setting priors for its transformation layers means defining a distribution over distributions. Our approach is to ensure that the flow stays close to its base distribution a priori, unless the data strongly suggests otherwise. We achieve this by constraining the scaling and shifting transformations using the bounded tanh function:

$$
\begin{aligned}
g_{\text{scale}}(\alpha^{(i)}) &= \alpha_{\max}^{\tanh(\alpha^{(i)})} \\
g_{\text{shift}}(\mu^{(i)}) &= \mu_{\max} \cdot \tanh(\mu^{(i)}),
\end{aligned}
\tag{9}
$$

where $\alpha_{\max}$ and $\mu_{\max}$ cap the maximum scaling and shifting transformation, preventing extreme deviations from the base distribution. When the flow parameters $\boldsymbol{\phi} = \mathbf{0}$, both $\alpha^{(i)}$ and $\mu^{(i)}$ are zero (Eq. 3), making $g_{\text{scale}}(\alpha^{(i)}) = 1$ and $g_{\text{shift}}(\mu^{(i)}) = 0$, thus yielding the identity transformation. We then place a Gaussian prior on the flow parameters, $p(\boldsymbol{\phi}) = \mathcal{N}(\boldsymbol{\phi}; \mathbf{0}, \sigma_{\boldsymbol{\phi}}^2 \mathbf{I})$, with $\sigma_{\boldsymbol{\phi}}$ chosen through *prior predictive checks* (see Section 4.1). $p(\boldsymbol{\phi})$, combined with our base distribution being moment-matched to the top observations, serves as a meaningful empirical prior that centers the flow in high-density regions of the target. Finally, we place an (improper) flat prior on the log normalization constant, $p(C) = 1$.

## 3.4. Annealed optimization

Fitting a flow to a complex unnormalized target density $p_{\text{target}}(\mathbf{x})$ via direct regression on observations $\boldsymbol{\Xi} = (\mathbf{X}, \mathbf{y}, \boldsymbol{\sigma}^2)$ can be challenging due to both the unknown log normalizing constant and potential gradient instabilities during optimization. We found that a more robust approach is to gradually fit the flow to an *annealed* (tempered) target across training iterations $t = 0, \ldots, T_{\max}$, using an inverse temperature parameter $\beta_t \in [0, 1]$. The tempered target takes the following form (see Figure 2 for an illustration):

$$
\widetilde{f}_{\beta_t}(\mathbf{x}) = (1 - \beta_t) \log p_0(\mathbf{x}) + \beta_t \log p_{\text{target}}(\mathbf{x}),
\tag{10}
$$

where $p_0(\mathbf{x})$ is the flow's base distribution. This formulation has two key advantages: first, since the base distribution is normalized, we know the true log normalizing constant $C$ is zero when $\beta_t = 0$. Second, by initializing the flow parameters near zero, the flow starts close to its base distribution $p_0$, providing a stable initialization point.

The tempered observations are defined as:

$$
\begin{aligned}
\widetilde{\mathbf{X}}_{\beta_t} &= \mathbf{X} \\
\widetilde{\mathbf{y}}_{\beta_t} &= (1 - \beta_t) \log p_0(\mathbf{X}) + \beta_t \mathbf{y} \\
\widetilde{\boldsymbol{\sigma}}_{\beta_t}^2 &= \max\left\{ \beta_t^2 \boldsymbol{\sigma}^2, \sigma_{\min}^2 \right\}
\end{aligned}
\tag{11}
$$

where $p_0(\mathbf{X}) = (p_0(\mathbf{x}_1), \ldots, p_0(\mathbf{x}_N))$ denotes the base distribution evaluated at all observed points. We increase the inverse temperature $\beta_t$ according to a tempering schedule increasing

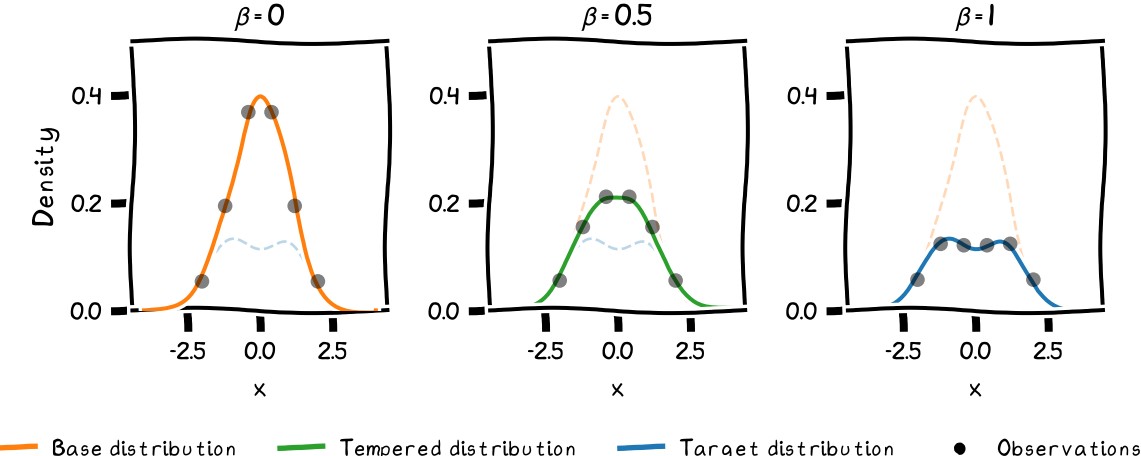

Figure 2: Annealed optimization strategy. The flow regression model is progressively fitted to a series of tempered observations, with the inverse temperature $\beta$ increasing over multiple training iterations, interpolating between the base and unnormalized target distributions.

from $\beta_0 = 0$ to $\beta_{t_{\mathrm{end}}} = 1$, where $t_{\mathrm{end}} \leq T_{\max}$ marks the end of tempering. After reaching $\beta = 1$, we can perform additional optimization iterations if needed. By default, we use a linear tempering schedule: $\beta_t = \beta_0 + \frac{t}{t_{\mathrm{end}}}(1 - \beta_0)$.

### 3.5. Normalizing flow regression algorithm

Having introduced the flow regression model and tempering approach, we now present the complete method in Algorithm 1, which returns the flow parameters $\phi$ and the log normalizing constant $C$. We follow a two-step approach: first, we fix the flow parameters $\phi$ and optimize the scalar parameter $C$ using, e.g., Brent's method (Brent, 1973), which is efficient as it requires only a single evaluation of the flow. Then, using this result as initialization, we jointly optimize both $\phi$ and $C$ with L-BFGS (Liu and Nocedal, 1989). Further details, including optimization termination criteria, are provided in Appendix A.1.

---

**Algorithm 1:** Normalizing Flow Regression

---

**Input:** Observations $(\mathbf{X}, \mathbf{y}, \boldsymbol{\sigma}^2)$, total number of tempered steps $t_{\mathrm{end}}$, maximum number of optimization iterations $T_{\max}$

**Output:** Flow $T_\phi$ approximating the target, log normalizing constant $C$

Compute and set the base distribution for the flow, using $(\mathbf{X}, \mathbf{y}, \boldsymbol{\sigma}^2)$ (Section 3.3);

**for** $t \leftarrow 0$ **to** $T_{max}$ **do**

    Set inverse temperature $\beta_t \in [0, 1]$ according to tempering schedule ($\beta_0 = 0$) ;

    Update tempered observations $(\widetilde{\mathbf{X}}_{\beta_t}, \widetilde{\mathbf{y}}_{\beta_t}, \widetilde{\boldsymbol{\sigma}}^2_{\beta_t})$ according to Eq. 11 ;

    Fix $\phi$ and optimize $C$ using fast 1$D$ optimization with objective in Eq. 6 ;

    Optimize $(\phi, C)$ jointly using L-BFGS with objective in Eq. 6 ;

**end**

---

## 4. Experiments

We evaluate our normalizing flow regression (NFR) method through a series of experiments. First, we conduct prior predictive checks to select our flow's prior settings (see Section 3.3). We then assess NFR's performance on both synthetic and real-world problems. For all the experiments, we use a masked autoregressive flow architecture and adopt the same fixed hyperparameters for the NFR algorithm (see Appendix A.1 for details).[4]

### 4.1. Prior predictive checks

As introduced in Section 3.3, we place a Gaussian prior $\mathcal{N}(\phi; \mathbf{0}, \sigma_\phi^2 \mathbf{I})$ on the flow parameters $\phi$. Since a normalizing flow represents a probability distribution, drawing parameters from this prior generates different realizations of possible distributions. We calibrate the prior variance $\sigma_\phi$ by visually inspecting these realizations, choosing a value that affords sufficient flexibility for the distributions to vary from the base distribution while maintaining reasonably smooth shapes.[5] Figure 3 shows density contours and samples from flow realizations under three different prior settings: $\sigma_\phi \in \{0.02, 0.2, 2\}$. Based on this analysis, we set the prior standard deviation $\sigma_\phi = 0.2$ for all subsequent experiments in the paper.

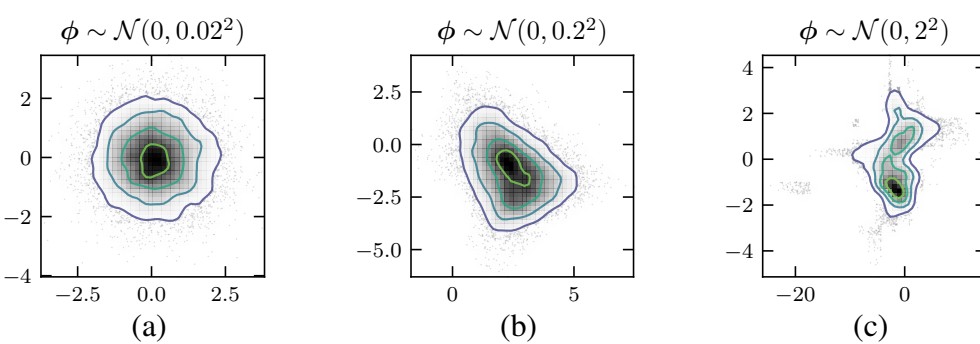

Figure 3: Effect of prior variance on normalizing flow behavior, using a standard Gaussian as the base distribution. The panels show flow realizations with different prior standard deviations $\sigma_\phi$: (a) The flow closely resembles the base distribution. (b) The flow exhibits controlled flexibility, allowing meaningful deviations while maintaining reasonable shapes. (c) The flow deviates significantly, producing complex and less plausible distributions.

### 4.2. Benchmark evaluations

We evaluate NFR on several synthetic and real-world problems, each defined by a *black-box* log-likelihood function and a log-prior function, or equivalently the target log-density function. The black-box nature of the likelihood means its gradients are unavailable, and we allow evaluations to be moderately expensive and potentially noisy. We are interested

---

4. The code implementation of NFR is available at github.com/acerbilab/normalizing-flow-regression.
5. This approach is a form of expert *prior elicitation* (Mikkola et al., 2024) about the expected shape of posterior distributions, leveraging our experience in statistical modeling.

in the offline inference setting, under the assumption that practitioners would have already performed multiple optimization runs for MAP estimation. Thus, to obtain training data for NFR, we collect log-density evaluations from MAP optimization runs using two popular black-box optimizers: CMA-ES (Hansen, 2016) and BADS, a hybrid Bayesian optimization method (Acerbi and Ma, 2017; Singh and Acerbi, 2024). For each problem, we allocate $3000D$ log-density evaluation where $D$ is the posterior dimension (number of model parameters). The details of the real-world problems are provided in Appendix A.3. For consistency, we present results from CMA-ES runs in the main text, with analogous BADS results and additional details in Appendix A.4. Example visualizations of the flow approximation and baselines are provided in Appendix A.9.

**Baselines.** We compare NFR against three baselines:

1. **Laplace approximation** (Laplace; MacKay, 2003), which constructs a Gaussian approximation using the MAP estimate and numerical computation of the Hessian, requiring additional log-density evaluations (Brodtkorb and D'Errico, 2022).

2. **Black-box variational inference** (BBVI; Ranganath et al., 2014), using the same normalizing flow architecture as NFR plus a learnable diagonal Gaussian base distribution. BBVI estimates ELBO gradients using the score function (REINFORCE) estimator with control variates, optimized using Adam (Kingma and Ba, 2014). We consider BBVI using both $3000D$ and $10 \times 3000D$ target density evaluations, with the latter being substantially more than NFR presented as a 'higher budget' baseline. Details on the implementation are provided in Appendix A.5.

3. **Variational sparse Bayesian quadrature** (VSBQ; Li et al., 2024), which like NFR uses existing evaluations to estimate the posterior. VSBQ fits a sparse Gaussian process to the log-density evaluations and runs variational inference on this surrogate with a Gaussian mixture model. We give VSBQ the same $3000D$ evaluations as NFR.

NFR and VSBQ are directly comparable as surrogate-based offline inference methods. BBVI requires additional evaluations of the target log density during training and is included as a strong online black-box inference baseline. Laplace requires additional log-density evaluation for the Hessian and serves as a popular approximate inference baseline.

**Metrics.** We assess algorithm performance by comparing the returned solutions against ground-truth posterior samples and normalizing constants. We use three metrics: the absolute difference between the true and estimated log normalizing constant ($\Delta$LML); the mean marginal total variation distance (MMTV); and the "Gaussianized" symmetrized KL divergence (GsKL) between the approximate and the true posterior (Acerbi, 2020; Li et al., 2024). MMTV quantifies discrepancies between marginals, while GsKL evaluates the overall joint distribution. Following previous recommendations, we consider approximations successful when $\Delta$LML $< 1$, MMTV $< 0.2$ and GsKL $< \frac{1}{8}$, with lower values indicating better performance (see Appendix A.2). For the stochastic methods (BBVI, VSBQ, and NFR), we report median performance and bootstrapped 95% confidence intervals from ten independent runs. We report only the median for the Laplace approximation, which is deterministic. Statistically significant best results are bolded, and metric values exceeding the desired thresholds are highlighted in red. See Appendix A.2 for further details.

### 4.2.1. SYNTHETIC PROBLEMS

**Multivariate Rosenbrock-Gaussian ($D = 6$).** We first test NFR on a six-dimensional synthetic target density with known complex geometry (Li et al., 2024). The target density takes the form:

$$p(\mathbf{x}) \propto e^{\mathcal{R}(x_1, x_2)} e^{\mathcal{R}(x_3, x_4)} \mathcal{N}([x_5, x_6]; \mathbf{0}, \mathbb{I}) \cdot \mathcal{N}(\mathbf{x}; \mathbf{0}, 3^2\mathbb{I}), \qquad (12)$$

which combines two exponentiated Rosenbrock ('banana') functions $\mathcal{R}(x, y)$ and a two-dimensional Gaussian density with an overall isotropic Gaussian prior.

From Figure 4 and Table 1, we see that both NFR and VSBQ perform well, with all metrics below the desired thresholds. Still, NFR consistently outperforms VSBQ across all metrics, achieving excellent approximation quality on this complex target. In contrast, BBVI suffers from slow convergence and potential local minima, with several metrics exceeding the thresholds even with a $10\times$ budget. Unsurprisingly, the Laplace approximation fails to capture the target's highly non-Gaussian structure.

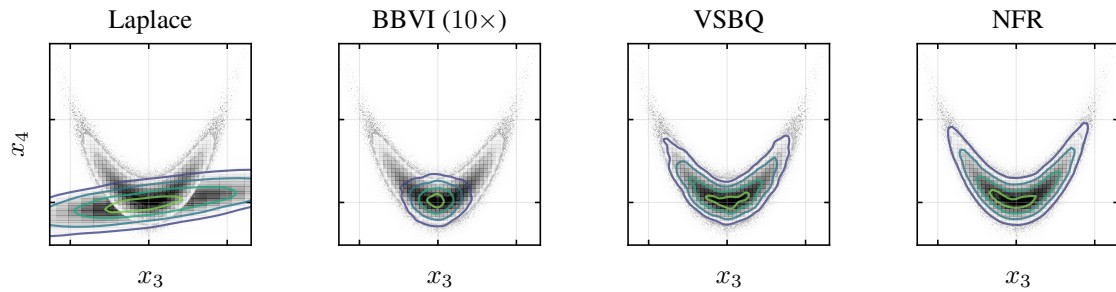

Figure 4: Multivariate Rosenbrock-Gaussian ($D = 6$). Example contours of the marginal density for $x_3$ and $x_4$, for different methods. Ground-truth samples are in gray.

Table 1: Multivariate Rosenbrock-Gaussian ($D = 6$).

|  | $\mathbf{\Delta LML}$ ($\downarrow$) | $\mathbf{MMTV}$ ($\downarrow$) | $\mathbf{GsKL}$ ($\downarrow$) |
|---|---|---|---|
| Laplace | 1.3 | 0.24 | 0.91 |
| BBVI ($1\times$) | 1.3 [1.2,1.4] | 0.23 [0.22,0.24] | 0.54 [0.52,0.56] |
| BBVI ($10\times$) | 1.0 [0.72,1.2] | 0.24 [0.19,0.25] | 0.46 [0.34,0.59] |
| VSBQ | 0.20 [0.20,0.20] | 0.037 [0.035,0.038] | 0.018 [0.017,0.018] |
| NFR | **0.013** [0.0079,0.017] | **0.028** [0.026,0.030] | **0.0042** [0.0024,0.0068] |

**Lumpy ($D = 10$).** Our second test uses a fixed instance of the *lumpy* distribution (Acerbi, 2018), a mildly multimodal density represented by a mixture of 12 partially overlapping multivariate Gaussian components in ten dimensions. For this target distribution, all methods except Laplace perform well with metrics below the target thresholds, and NFR again achieves the best performance. The Laplace approximation provides reasonable estimates

of the normalizing constant and marginal distributions but struggles with the full joint distribution. Further details are provided in Appendix A.4.

### 4.2.2. Real-world problems

**Bayesian timing model ($D = 5$).** Our first real-world application comes from cognitive neuroscience, where Bayesian observer models are applied to explain human time perception (Jazayeri and Shadlen, 2010; Acerbi et al., 2012; Acerbi, 2020). These models assume that participants in psychophysical experiments are themselves performing Bayesian inference over properties of sensory stimuli (e.g., duration), using Bayesian decision theory to generate percept responses (Pouget et al., 2013; Ma et al., 2023). To make the inference scenario more challenging and realistic, we include log-likelihood estimation noise with $\sigma_n = 3$, similar to what practitioners would find if estimating the log likelihood via Monte Carlo instead of precise numerical integration methods (van Opheusden et al., 2020).

As shown in Table 2, NFR and VSBQ accurately approximate this posterior, while BBVI (10×) shows slightly worse performance with larger confidence intervals. BBVI (1×) fails to converge, with all metrics exceeding the thresholds. The Laplace approximation is not applicable here due to the likelihood noise preventing reliable numerical differentiation.

Table 2: Bayesian timing model ($D = 5$).

|  | **ΔLML** (↓) | **MMTV** (↓) | **GsKL** (↓) |
|---|---|---|---|
| BBVI (1×) | 1.6 [1.1,2.5] | 0.29 [0.27,0.34] | 0.77 [0.67,1.0] |
| BBVI (10×) | **0.32** [0.036,0.66] | 0.11 [0.088,0.15] | 0.13 [0.052,0.23] |
| VSBQ | **0.21** [0.18,0.22] | **0.044** [0.039,0.049] | **0.0065** [0.0059,0.0084] |
| NFR | **0.18** [0.17,0.24] | **0.049** [0.041,0.052] | **0.0086** [0.0053,0.011] |

**Lotka-Volterra model ($D = 8$).** Our second real-world test examines parameter inference for the Lotka-Volterra predatory-prey model (Carpenter, 2018), a classic system of coupled differential equations that describe population dynamics. Using data from Howard (2009), we infer eight parameters governing the interaction rates, initial population sizes, and observation noise levels.

Table 3 shows that NFR significantly outperforms all baselines on this problem. BBVI, VSBQ, and the Laplace approximation achieve acceptable performance, with all metrics below the desired thresholds except for the GsKL metric in the Laplace approximation.

**Bayesian causal inference in multisensory perception ($D = 12$).** Our final and most challenging test examines a model of multisensory perception from computational neuroscience (Acerbi et al., 2018). The model describes how humans decide whether visual and vestibular (balance) sensory cues share a common cause – a fundamental problem in neural computation (Körding et al., 2007). The model's likelihood is mildly expensive ($> 3$s per evaluation), due to the numerical integration required for its computation.

The high dimensionality and complex likelihood of this model make it particularly challenging for several methods. Due to a non-positive-definite numerical Hessian, the Laplace

Table 3: Lotka-Volterra model ($D = 8$).

|  | $\Delta$LML ($\downarrow$) | MMTV ($\downarrow$) | GsKL ($\downarrow$) |
|---|---|---|---|
| Laplace | 0.62 | 0.11 | 0.14 |
| BBVI (1×) | 0.47 [0.42,0.59] | 0.055 [0.048,0.063] | 0.029 [0.025,0.034] |
| BBVI (10×) | 0.24 [0.23,0.36] | 0.029 [0.025,0.039] | 0.0087 [0.0052,0.014] |
| VSBQ | 0.95 [0.93,0.97] | 0.085 [0.084,0.089] | 0.060 [0.059,0.062] |
| NFR | **0.18** [0.17,0.18] | **0.016** [0.015,0.017] | **0.00066** [0.00056,0.00083] |

approximation is inapplicable. The likelihood's computational cost makes BBVI (1×), let alone 10×, impractical to benchmark and to use in practice.[6] Thus, we focus on comparing NFR and VSBQ (Table 4). NFR performs remarkably well on this challenging posterior, with metrics near or just above our desired thresholds, while VSBQ fails to produce a usable approximation.

Table 4: Multisensory ($D = 12$).

|  | $\Delta$LML ($\downarrow$) | MMTV ($\downarrow$) | GsKL ($\downarrow$) |
|---|---|---|---|
| VSBQ | 4.1e+2 [3.0e+2,5.4e+2] | 0.87 [0.82,0.93] | 2.0e+2 [1.1e+2,4.1e+4] |
| NFR | **0.82** [0.75,0.90] | **0.13** [0.12,0.14] | **0.11** [0.091,0.16] |

## 5. Discussion

In this paper, we introduced normalizing flow regression as a novel method for performing approximate Bayesian posterior inference, using offline likelihood evaluations. Normalizing flows offer several advantages: they ensure proper probability distributions, enable easy sampling, scale efficiently with the number of likelihood evaluations, and can flexibly incorporate prior knowledge of posterior structure. While we demonstrated that our proposed approach works well, it has limitations which we discuss in Appendix A.6. For practitioners, we further provide an ablation study of our design choices in Appendix A.7 and a discussion on diagnostics for detecting potential failures in the flow approximation in Appendix A.8.

In this work, we focused on using log-density evaluations from MAP optimization due to its widespread practice, but our framework can be extended to incorporate other likelihood evaluation sources. For example, it could include evaluations of pre-selected plausible parameter values, as seen in cosmology (Rizzato and Sellentin, 2023), or actively and sequentially acquire new evaluations based on the current posterior estimate (Acerbi, 2018; Greenberg et al., 2019). We leave these topics as future work.

---

6. From partial runs, we estimated $> 100$ hours *per run* for BBVI (1×) on our computing setup.

## Acknowledgments

This work was supported by Research Council of Finland (grants 358980 and 356498), and by the Flagship programme: Finnish Center for Artificial Intelligence FCAI. The authors wish to thank the Finnish Computing Competence Infrastructure (FCCI) for supporting this project with computational and data storage resources.

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

## Appendix A.

This appendix provides additional details and analyses to complement the main text, included in the following sections:

- Normalizing flow regression algorithm details, A.1

- Metrics description, A.2

- Real-world problems description, A.3

- Additional experimental results, A.4

- Black-box variational inference implementation, A.5

- Limitations, A.6

- Ablation studies, A.7

- Diagnostics, A.8

- Visualization of posteriors, A.9

### A.1. Normalizing flow regression algorithm details

**Inference space.** NFR, VSBQ, Laplace approximation, and BBVI all operate in an *unbounded* parameter space, which we call the *inference space*. Originally bounded parameters are first mapped to the inference space and then rescaled and shifted based on user-specified plausible ranges, such as the 68.2% percentile interval of the prior. After transformation, the plausible ranges in the inference space are standardized to $[-0.5, 0.5]$. An appropriate Jacobian correction is applied to the log-density values in the inference space. Similar transformations are commonly used in probabilistic inference software (Carpenter et al., 2017; Huggins et al., 2023). The approximate posterior samples are transformed back to the original space via the inverse transform for performance evaluation against the ground truth posterior samples.

**Noise shaping hyperparameter choice for NFR.** The function $s(\cdot)$ in Eq. 8 acts as a noise shaping mechanism that increases observation uncertainty for lower-density regions, further preventing overfitting to low-density observations (Li et al., 2024). It is worth noting that the noise shaping mechanism introduces artificial noise even when the density is measured exactly, and this is a feature of the algorithm designed to reduce the undesired influence of low-density observations. We define $s(\cdot)$ as a piecewise linear function,

$$s(f_{\max} - f_n) = \begin{cases} 0 & \text{if } f_{\max} - f_n < \delta_1, \\ \lambda(f_{\max} - f_n - \delta_1) & \text{if } \delta_1 \leq f_{\max} - f_n \leq \delta_2, \\ \lambda(\delta_2 - \delta_1) & \text{if } f_{\max} - f_n > \delta_2. \end{cases} \quad \text{(A.1)}$$

Here, $\delta_1$ and $\delta_2$ define the thresholds for moderate and extremely low log-density values, respectively. In practice, we approximate the unknown difference $f_{\max} - f_n$ with $y_{\max} - y_n$, where $y_{\max} = \max_n y_n$ is the maximum observed log-density value. We set $y_{\text{low}} = \max_n(y_n -$

$1.96\sigma_n) - \delta_2$ for Eq. 8. For all problems, we set $\lambda = 0.05$ following Li et al. (2024). The thresholds for moderate density and extremely low density are defined as $\delta_1 = 10D$, $\delta_2 = 50D$, where $D$ is the target posterior dimension.[7] The extremely low-density value is computed as $y_{\text{low}} = \max_n(y_n - 1.96\sigma_n) - \delta_2$.

**Normalizing flow architecture specifications.** For all experiments, we use the masked autoregressive flow (MAF; Papamakarios et al., 2017) with the original implementation from Durkan et al. (2020). The flow consists of 11 transformation layers, each comprising an affine autoregressive transform followed by a reverse permutation transform. As described in Section 3.3, the flow's base distribution is a diagonal multivariate Gaussian estimated from observations with sufficiently high log-density values. Specifically, we select observations satisfying $y_n - 1.96\sigma_n \geq \delta_1$ and compute the mean and covariance directly from these selected points $\mathbf{x}_n$. The maximum scaling factor $\alpha_{\max}$ and $\mu_{\max}$ are chosen such that the normalizing flow exhibits controlled flexibility from the base distribution, as illustrated in Section 4.1. We set $\alpha_{\max} = 1.5$ and $\mu_{\max} = 1$ (Eq. 9) across the experiments.

**Initialization of regression model parameters.** The parameter set for the normalizing flow regression model is $\boldsymbol{\psi} = (\boldsymbol{\phi}, C)$, where $\boldsymbol{\phi}$ represents the flow parameters, i.e., the parameters of the neural networks. We initialize $\boldsymbol{\phi}$ by multiplying the default Py-Torch initialization (Paszke et al., 2019) by $10^{-3}$ to ensure the flow starts close to its base distribution. The parameter $C$ is initialized to zero.

**Termination criteria for normalizing flow regression.** For all problems, we set the number of annealed steps $t_{\text{end}} = 20$ and the maximum number of training iterations $T_{\max} = 30$. At each training iteration, the L-BFGS optimizer is run with a maximum of 500 iterations and up to 2000 function evaluations. The L-BFGS optimization terminates if the directional derivative falls below a threshold of $10^{-5}$ or if the maximum absolute change in the loss function over five consecutive iterations is less than $10^{-5}$.

**Training dataset.** For each benchmark problem, MAP estimation is performed to find the target posterior mode. We launch MAP optimization runs from random initial points and collect multiple optimization traces as the training dataset for NFR and VSBQ. The total number of target density evaluations is fixed to $3000D$. It is worth noting that the MAP estimate depends on the choice of parameterization. We align with the practical usage scenario where optimization is performed in the original parameter space and the parameter bounds are dealt with by the optimizers (in our case, CMA-ES and BADS).

## A.2. Metrics description

Following Acerbi (2020); Li et al. (2024), we use three metrics: the absolute difference $\Delta$LML between the true and estimated log normalizing constant (log marginal likelihood); the mean marginal total variation distance (MMTV); and the "Gaussianized" symmetrized KL divergence (GsKL) between the approximate and true posterior. For each problem,

---

7. El Gammal et al. (2023); Li et al. (2024) set the low-density thresholds by referring to the log-density range of a standard $D$-dimensional multivariate Gaussian distribution, which requires computing an inverse CDF of a chi-squared distribution. However, this computation for determining the extremely low-density threshold can numerically overflow to $\infty$. Therefore, we use a linear approximation in $D$, similar to Huggins et al. (2023).

ground-truth posterior samples are obtained through rejection sampling, extensive MCMC, or analytical/numerical methods. The ground-truth log normalizing constant is computed analytically, using numerical quadrature methods, or estimated from posterior samples via Geyer's reverse logistic regression (Geyer, 1994). For completeness, we describe below the metrics and desired thresholds in detail, largely following Li et al. (2024):

- $\Delta$LML measures the absolute difference between true and estimated log marginal likelihood. We aim for an LML loss $< 1$, as differences in log model evidence $\ll 1$ are considered negligible for model selection (Burnham and Anderson, 2003).

- The MMTV quantifies the (lack of) overlap between true and approximate posterior marginals, defined as

$$\text{MMTV}(p, q) = \sum_{d=1}^{D} \int_{-\infty}^{\infty} \frac{\left|p_d^{\text{M}}(x_d) - q_d^{\text{M}}(x_d)\right|}{2D} dx_d, \qquad (A.2)$$

where $p_d^{\text{M}}$ and $q_d^{\text{M}}$ denote the marginal densities of $p$ and $q$ along the $d$-th dimension. An MMTV metric of 0.2 indicates that, on average across dimensions, the posterior marginals have an 80% overlap. As a rule of thumb, we consider this level of overlap (MMTV $< 0.2$) as the threshold for a reasonable posterior approximation.

- The (averaged) GsKL metric evaluates differences in means and covariances:

$$\text{GsKL}(p, q) = \frac{1}{2D} \left[D_{\text{KL}}\left(\mathcal{N}[p]||\mathcal{N}[q]\right) + D_{\text{KL}}(\mathcal{N}[q]||\mathcal{N}[p])\right], \qquad (A.3)$$

where $D_{\text{KL}}\left(p||q\right)$ is the Kullback-Leibler divergence between distributions $p$ and $q$ and $\mathcal{N}[p]$ denotes a multivariate Gaussian with the same mean and covariance as $p$ (similarly for $q$). This metric has a closed-form expression in terms of means and covariance matrices. For reference, two Gaussians with unit variance whose means differ by $\sqrt{2}$ (resp. $\frac{1}{2}$) yield GsKL values of 1 (resp. $\frac{1}{8}$). As a rule of thumb, we consider GsKL $< \frac{1}{8}$ to indicate a sufficiently accurate posterior approximation.

## A.3. Real-world problems description

**Bayesian timing model ($D = 5$).** We analyze data from a sensorimotor timing experiment in which participants were asked to reproduce time intervals $\tau$ between a mouse click and screen flash, with $\tau \sim \text{Uniform}[0.6, 0.975]$ s (Acerbi et al., 2012). The model assumes participants receive noisy sensory measurements $t_s \sim \mathcal{N}(\tau, w_s^2 \tau^2)$ and they generate an estimate $\tau_\star$ by combining this sensory evidence with a Gaussian prior $\mathcal{N}\left(\tau; \mu_p, \sigma_p^2\right)$ and taking the posterior mean. Their reproduced times then include motor noise, $t_m \sim \mathcal{N}(\tau_\star, w_m^2 \tau_\star^2)$, and each trial has probability $\lambda$ of a "lapse" (e.g., misclick) yielding instead $t_m \sim \text{Uniform}[0, 2]$ s. The model has five parameters $\boldsymbol{\theta} = (w_s, w_m, \mu_p, \sigma_p, \lambda)$, where $w_s$ and $w_m$ are Weber fractions quantifying perceptual and motor variability. We adopt a spline-trapezoidal prior for all parameters. The spline-trapezoidal prior is uniform between the plausible ranges of the parameter while falling smoothly as a cubic spline to zero toward the parameter bounds. We infer the posterior for a representative participant from Acerbi et al. (2012). As explained in the main text, we make the inference scenario more challenging and realistic by

including log-likelihood estimation noise with $\sigma_n = 3$. This noise magnitude is analogous to what practitioners would find by estimating the log-likelihood via Monte Carlo instead of using numerical integration methods (van Opheusden et al., 2020).

**Lotka-Volterra model ($D = 8$).** The model describes population dynamics through coupled differential equations:

$$\frac{\mathrm{d}u}{\mathrm{d}t} = \alpha u - \beta uv; \quad \frac{\mathrm{d}v}{\mathrm{d}t} = -\gamma v + \delta uv;$$

where $u(t)$ and $v(t)$ represent prey and predator populations at time $t$, respectively. Using data from Howard (2009), we infer eight parameters: four rate constants ($\alpha$, $\beta$, $\gamma$, $\delta$), initial conditions ($u(0)$, $v(0)$), and observation noise intensities ($\sigma_u$, $\sigma_v$). The likelihood is computed by solving the equations numerically using the Runge–Kutta method. See Carpenter (2018) for further details of priors and model implementations.

**Bayesian causal inference in multisensory perception ($D = 12$).** In the experiment, participants seated in a moving chair judged whether the direction of their motion $s_{\text{vest}}$ matched that of a visual stimulus $s_{\text{vis}}$ ('same' or 'different'). The model assumes participants receive noisy measurements $z_{\text{vest}} \sim \mathcal{N}(s_{\text{vest}}, \sigma_{\text{vest}}^2)$ and $z_{\text{vis}} \sim \mathcal{N}(s_{\text{vis}}, \sigma_{\text{vis}}^2(c))$, where $\sigma_{\text{vest}}$ is vestibular noise and $\sigma_{\text{vis}}(c)$ represents visual noise under three different coherence levels $c$. Each sensory noise parameter includes both a base standard deviation and a Weber fraction scaling factor. The Bayesian causal inference observer model also incorporates a Gaussian spatial prior, probability of common cause, and lapse rate for random responses, totaling 12 parameters. The model's likelihood is mildly expensive ($\sim$ 3s per evaluation), due to numerical integration used to compute the observer's posterior over causes, which would determine their response in each trial ('same' or 'different'). We adopt a spline-trapezoidal prior for all parameters, which remains uniform within the plausible parameter range and falls smoothly to zero near the bounds using a cubic spline. We fit the data of representative subject S11 from Acerbi et al. (2018).

### A.4. Additional experimental results

**Lumpy distribution ($D = 10$).** Table A.1 presents the results for the ten-dimensional lumpy distribution, omitted from the main text due to space constraints. All methods, except Laplace, achieve metrics below the target thresholds, with NFR performing best. While the Laplace approximation provides reasonable estimates of the normalizing constant and marginal distributions, it struggles with the full joint distribution.

**Results from MAP runs with BADS optimizer.** We present here the results of applying NFR and VSBQ to the MAP optimization traces from the BADS optimizer (Acerbi and Ma, 2017), instead of CMA-ES used in the main text. BADS is an efficient hybrid Bayesian optimization method that also deals with noisy observations like CMA-ES. The results for the other baselines (BBVI, Laplace) are the same as those reported in the main text, since these methods do not reuse existing (offline) optimization traces, but we repeat them here for ease of comparison.

The full results are shown in Table A.2, A.3, A.4, A.5, and A.6. From the tables, we can see that NFR still achieves the best performance for all problems. For the challenging 12D

Table A.1: Lumpy ($D = 10$).

| | $\Delta$LML ($\downarrow$) | MMTV ($\downarrow$) | GsKL ($\downarrow$) |
|---|---|---|---|
| Laplace | 0.81 | 0.15 | 0.22 |
| BBVI (1×) | 0.42 [0.40,0.51] | 0.065 [0.061,0.079] | 0.029 [0.023,0.035] |
| BBVI (10×) | 0.32 [0.28,0.41] | 0.046 [0.041,0.051] | 0.013 [0.0095,0.015] |
| VSBQ | 0.11 [0.097,0.15] | 0.033 [0.031,0.038] | 0.0070 [0.0066,0.0090] |
| NFR | **0.026** [0.016,0.040] | **0.022** [0.022,0.024] | **0.0020** [0.0018,0.0023] |

multisensory problem, the metrics $\Delta$LML and GsKL slightly exceed the desired thresholds. Additionally, as shown by comparing Table 4 in the main text and Table A.6, NFR performs slightly worse when using evaluations from BADS, compared to CMA-ES. We hypothesize that this is because BADS converges rapidly to the posterior mode, resulting in less evaluation coverage on the posterior log-density function, as also noted by Li et al. (2024). In sum, our results about the accuracy of NFR qualitatively hold regardless of the optimizer.

Table A.2: Multivariate Rosenbrock-Gaussian ($D = 6$). (BADS)

| | $\Delta$LML ($\downarrow$) | MMTV ($\downarrow$) | GsKL ($\downarrow$) |
|---|---|---|---|
| Laplace | 1.3 | 0.24 | 0.91 |
| BBVI (1×) | 1.3 [1.2,1.4] | 0.23 [0.22,0.24] | 0.54 [0.52,0.56] |
| BBVI (10×) | 1.0 [0.72,1.2] | 0.24 [0.19,0.25] | 0.46 [0.34,0.59] |
| VSBQ | 0.19 [0.19,0.20] | 0.038 [0.037,0.039] | 0.018 [0.017,0.018] |
| NFR | **0.0067** [0.0031,0.012] | **0.028** [0.026,0.031] | **0.0053** [0.0032,0.0060] |

Table A.3: Lumpy. (BADS)

| | $\Delta$LML ($\downarrow$) | MMTV ($\downarrow$) | GsKL ($\downarrow$) |
|---|---|---|---|
| Laplace | 0.81 | 0.15 | 0.22 |
| BBVI (1×) | 0.42 [0.40,0.51] | 0.065 [0.061,0.079] | 0.029 [0.023,0.035] |
| BBVI (10×) | 0.32 [0.28,0.41] | 0.046 [0.041,0.051] | 0.013 [0.0095,0.015] |
| VSBQ | **0.029** [0.0099,0.043] | 0.034 [0.033,0.037] | 0.0065 [0.0060,0.0073] |
| NFR | 0.072 [0.057,0.087] | **0.029** [0.028,0.031] | **0.0021** [0.0017,0.0026] |

**Runtime analysis.** To assess computational efficiency, we reran each method—BBVI (with 1× budget, 10 Monte Carlo samples, learning rate 0.001), VSBQ, and NFR—five

Table A.4: Bayesian timing model. (BADS)

|  | $\Delta$LML ($\downarrow$) | MMTV ($\downarrow$) | GsKL ($\downarrow$) |
|---|---|---|---|
| BBVI (1×) | 1.6 [1.1,2.5] | 0.29 [0.27,0.34] | 0.77 [0.67,1.0] |
| BBVI (10×) | **0.32** [0.036,0.66] | 0.11 [0.088,0.15] | 0.13 [0.052,0.23] |
| VSBQ | **0.22** [0.18,0.42] | **0.057** [0.045,0.074] | **0.010** [0.0070,0.14] |
| NFR | **0.24** [0.21,0.27] | **0.060** [0.052,0.076] | **0.014** [0.0088,0.017] |

Table A.5: Lotka-volterra model. (BADS)

|  | $\Delta$LML ($\downarrow$) | MMTV ($\downarrow$) | GsKL ($\downarrow$) |
|---|---|---|---|
| Laplace | 0.62 | 0.11 | 0.14 |
| BBVI (1×) | 0.47 [0.42,0.59] | 0.055 [0.048,0.063] | 0.029 [0.025,0.034] |
| BBVI (10×) | 0.24 [0.23,0.36] | 0.029 [0.025,0.039] | 0.0087 [0.0052,0.014] |
| VSBQ | 1.0 [1.0,1.0] | 0.084 [0.081,0.087] | 0.063 [0.061,0.064] |
| NFR | **0.18** [0.17,0.18] | **0.015** [0.014,0.016] | **0.00074** [0.00057,0.00092] |

Table A.6: Multisensory. (BADS)

|  | $\Delta$LML ($\downarrow$) | MMTV ($\downarrow$) | GsKL ($\downarrow$) |
|---|---|---|---|
| VSBQ | 1.5e+3 [6.2e+2,2.1e+3] | 0.87 [0.81,0.90] | 1.2e+4 [2.0e+2,1.4e+8] |
| NFR | **1.1** [0.95,1.3] | **0.15** [0.13,0.19] | **0.22** [0.15,0.94] |

times independently on an NVIDIA V100 GPU for each problem. Table A.7 reports the average runtimes (in seconds) along with standard deviations.

The Laplace approximation is generally the fastest approach, except in cases involving expensive likelihood evaluations. BBVI (1×) is fast for models with cheap likelihood evaluations, but becomes computationally demanding or infeasible for models with expensive likelihoods. The runtime of BBVI (10×) is approximately ten times that of BBVI (1×). NFR's runtime is significantly influenced by the number of annealing steps; we used 20 steps across all experiments. However, for several problems, NFR performs comparably well with fewer or even no annealing steps (see Appendix A.7), potentially enabling substantial speed-ups. We defer a more aggressive optimization of the NFR pipeline to future work.

## A.5. Black-box variational inference implementation

**Normalizing flow architecture specifications and initialization.** For BBVI, we use the same normalizing flow architecture as in NFR. The base distribution of the normalizing

Table A.7: Average runtime (in seconds) across five runs for each method.

| Model | Laplace | BBVI (1x) | VSBQ | NFR |
|---|---|---|---|---|
| Rosenbrock-Gaussian (D=6) | $\sim$1 | $171 \pm 10$ | $391 \pm 33$ | $1374 \pm 27$ |
| Lumpy (D=10) | $\sim$1 | $383 \pm 24$ | $949 \pm 49$ | $1499 \pm 70$ |
| Noisy timing (D=5) | N/A | $1167 \pm 85$ | $301 \pm 20$ | $937 \pm 57$ |
| Lotka-Volterra (D=8) | $\sim$1 | $295 \pm 6$ | $817 \pm 151$ | $1384 \pm 38$ |
| Multisensory (D=12) | $\sim$3 hr | $>$30 hr | $1345 \pm 185$ | $1742 \pm 11$ |

flow is set to a learnable diagonal multivariate Gaussian, unlike in NFR where the means and variances can be estimated from the MAP optimization runs. The base distribution is initialized as a multivariate Gaussian with mean zero and standard deviations set to one-tenth of the plausible ranges. The transformation layers, parameterized by neural networks, are initialized using the same procedure as in NFR (see Appendix A.1).

**Stochastic optimization.** As described in the main text, BBVI is performed by optimizing the ELBO using the Adam optimizer. To give BBVI the best chance of performing well, for each problem we conducted a grid search over the learning rate $\{0.01, 0.001\}$ and the number of Monte Carlo samples for gradient estimation $\{1, 10, 100\}$, selecting the best-performing configuration based on the estimated ELBO value and reporting the performance metrics accordingly. Following Li et al. (2024), we further apply a control variate technique to reduce the variance of the ELBO gradient estimator.

## A.6. Limitations

In this work, we leverage normalizing flows as a regression surrogate to approximate the log-density function of a probability distribution. This methodology inherits the limitations of surrogate modeling approaches. Regardless of the source, the training dataset needs to sufficiently cover regions of non-negligible probability mass. In high-dimensional settings, this implies that the required number of training points grows exponentially, eventually becoming impractical (Li et al., 2024). In practice, similarly to other surrogate-based methods, we expect our method to be applicable to models with up to 10-15 parameters, as demonstrated by the 12-dimensional example in the main text. Scalability beyond $D \approx 20$ remains to be investigated.

In the paper, we focus on obtaining training data from MAP optimization traces. In this case, care must be taken to ensure the MAP estimate does not fall exactly on parameter bounds; otherwise, transformations into inference space (Appendix A.1) could push log-density observations to infinity, rendering them uninformative for constructing the normalizing flow surrogate. This issue is an old and well-known problem in approximate Bayesian inference (e.g., for the Laplace approximation, MacKay, 1998) and can be mitigated by imposing priors that vanish at the bounds (Gelman et al., 2013, Chapter 13), such as the spline-trapezoidal prior as in Appendix A.3). Additionally, fitting a regression model to pointwise log-density observations may become less meaningful in certain scenarios, e.g., when the likelihood is unbounded or highly non-smooth.

Our proposed technique, normalizing flow regression, jointly estimates both the flow parameters and the normalizing constant. The latter is a notoriously challenging quantity to infer even when target distribution samples are available (Geyer, 1994; Gutmann and Hyvärinen, 2010; Gronau et al., 2017). We impose priors over the flow for mitigating the non-identifiability issue (Section 3.3) and further apply an annealed optimization technique (Section 3.4), which we empirically find improves the posterior approximation and normalizing constant estimation (Section 4, Appendix A.7). Compared to Gaussian process surrogates, where smoothness is explicitly controlled by the covariance kernel, our priors over the flow are more implicit in governing the regularity of the (log-)density function, yet more explicit in shaping the overall distribution. Nevertheless, these strategies are not silver bullets, and we strongly recommend performing diagnostic checks on the flow approximation (Appendix A.8) whenever possible.

Finally, in this paper, our focus is on problems with relatively smooth, unimodal or mildly multimodal posteriors, which are common in real-world statistical modeling. Normalizing flows are known to struggle when the base distribution and target distribution exhibit significant topological differences (Cornish et al., 2020; Stimper et al., 2022). In the density estimation context—where the flow is trained via maximum likelihood on samples from the target distribution, a substantially easier setting than ours—there exist specialized approaches to improve performance for multimodal distributions (Stimper et al., 2022) and distributions with a mix of light and heavy tails (Amiri et al., 2024). A detailed investigation into handling such challenging posterior structures in regression settings and potential extensions is left for future research.

## A.7. Ablation studies

To validate our key design choices, we conducted ablation studies examining three components of NFR: the likelihood function (Section 3.2), flow priors (Section 3.3), and annealed optimization (Section 3.4). We tested these using two problems from our benchmark: the Bayesian timing model ($D = 5$) and the challenging multisensory perception model ($D = 12$). As shown in Table A.8, our proposed combination of Tobit likelihood, flow prior settings, and annealed optimization achieves the best overall performance. The progression of results reveals several insights.

First, noise shaping in the regression likelihood proves crucial. The basic Gaussian observation noise without noise shaping, as defined in Eq. 7, yields poor approximations of the true target posterior. Adding noise shaping to the regression likelihood significantly improves performance. Switching then to our Tobit likelihood (Eq. 8) provides marginally further benefits. Indeed, the Gaussian likelihood with noise shaping is a special case of the Tobit likelihood where the low-density threshold $y_{\text{low}}$ approaches negative infinity.

Second, the importance of annealing depends on problem complexity. While the low-dimensional timing model performs adequately without annealing, the 12-dimensional multisensory model requires it for stable optimization. This suggests annealing becomes crucial as dimensionality increases.

Finally, flow priors prove essential for numerical stability and performance. Without them, many optimization runs fail due to numerical errors (marked with asterisks in Table A.8), and even successful runs show substantially degraded performance.

Table A.8: Ablation experiments. The abbreviation 'ns' refers to noise shaping (Eq. A.1). Results marked with * indicate that multiple runs failed due to numerical errors.

| Ablation settings | | | Bayesian timing model ($D = 5$) | | | Multisensory ($D = 12$) | | |
|---|---|---|---|---|---|---|---|---|
| likelihood | with flow priors | annealing | $\Delta$LML | MMTV | GsKL | $\Delta$LML | MMTV | GsKL |
| Gaussian w/o ns | ✓ | ✓ | **0.16** [0.089,0.29] | 0.21 [0.18,0.30] | 0.42 [0.24,0.83] | 4.0 [1.9,7.1] | 0.44 [0.40,0.51] | 5.9 [3.4,9.3] |
| Gaussian w/ ns | ✓ | ✓ | **0.20** [0.18,0.23] | **0.055** [0.043,0.059] | **0.0096** [0.0074,0.013] | **0.87** [0.69,1.0] | **0.13** [0.11,0.15] | **0.12** [0.086,0.17] |
| Tobit | ✓ | ✗ | **0.20** [0.17,0.23] | **0.048** [0.044,0.052] | **0.0098** [0.0062,0.011] | 24. [18.,42.] | 0.82 [0.76,0.84] | 2.8e+2 [62.,9.0e+2] |
| Tobit | ✗ | ✓ | 6.7* [6.0,7.9] | 0.99* [0.99,1.0] | 2.6e+3* [1.6e+3,4.6e+3] | 0.86* [0.73,0.96] | 0.14* [0.13,0.17] | 0.25* [0.14,3.6] |
| Tobit | ✓ | ✓ | **0.18** [0.17,0.24] | **0.049** [0.041,0.052] | **0.0086** [0.0053,0.011] | **0.82** [0.75,0.90] | **0.13** [0.12,0.14] | **0.11** [0.091,0.16] |

## A.8. Diagnostics

When approximating a posterior through regression on a set of log-density evaluations, several issues can lead to poor-quality approximations. The training points may inadequately cover the true target posterior, and while the normalizing flow can extrapolate to missing regions, its accuracy in these areas is not guaranteed. Additionally, since we treat the unknown log normalizing constant $C$ as an optimization parameter, biased estimates can cause problems: overestimation leads to a hallucination of probability mass in low-density regions, while underestimation results in overly concentrated, mode-seeking behavior.

Given these potential issues, we recommend two complementary diagnostic approaches to practitioners to assess the quality of the flow approximation in addition to standard posterior predictive checks.

1. When additional noiseless target posterior density evaluations are available, we can use the fitted flow as a proposal distribution for Pareto smoothed importance sampling (PSIS; Vehtari et al., 2024). PSIS computes a Pareto $\hat{k}$ statistic that quantifies how well the proposal (the flow) approximates the target posterior. A value of $\hat{k} \leq 0.7$ indicates a good approximation, while $\hat{k} > 0.7$ suggests poor alignment with the posterior (Yao et al., 2018; Dhaka et al., 2021; Vehtari et al., 2024).[8] The target log density evaluations needed for this diagnostic can be computed in parallel for efficiency.

2. A simple yet effective complementary diagnostic approach uses *corner* plots (Foreman-Mackey, 2016) to visualize flow samples with pairwise two-dimensional marginal densities, alongside log-density observation points **X**. This visualization can reveal a

---

8. Apart from being a diagnostic, importance sampling can help refine the approximate posterior when $\hat{k} < 0.7$.

Table A.9: PSIS diagnostics. Both the median and the 95% confidence interval (CI) of the median are provided. We show the PSIS-$\hat{k}$ statistic computed with 100, 1000, and 2000 proposal samples. $\hat{k} > 0.7$ indicates potential issues and is reported in red. (CMA-ES)

| Problem | PSIS-$\hat{k}$ (100) | PSIS-$\hat{k}$ (1000) | PSIS-$\hat{k}$ (2000) |
|---|---|---|---|
| Multivariate Rosenbrock-Gaussian | 0.64 [0.38,0.87] | 0.88 [0.63,1.2] | 0.91 [0.75,1.0] |
| Lumpy | 0.36 [0.26,0.45] | 0.34 [0.30,0.42] | 0.39 [0.26,0.45] |
| Lotka-Volterra model | 0.50 [0.24,0.57] | 0.41 [0.27,0.52] | 0.39 [0.28,0.56] |
| Multisensory | 0.23 [0.15,0.50] | 0.37 [0.31,0.50] | 0.53 [0.43,0.56] |

common failure mode known as *hallucination* (De Souza et al., 2022; Li et al., 2024), where the surrogate model, the flow in our case, erroneously places significant probability mass in regions far from the training points.

We illustrate these two diagnostics in detail with examples below. For PSIS, we use the normalizing flow $q_\phi$ as the proposal distribution for importance sampling and compute the importance weights,

$$r_s = \frac{p_{\text{target}}(\mathbf{x}_s)}{q_\phi(\mathbf{x}_s)}, \quad \mathbf{x}_s \sim q_\phi(\mathbf{x}) \tag{A.4}$$

PSIS fits a generalized Pareto distribution using the importance ratios $r_s$ and returns the estimated shape parameter $\hat{k}$ which serves as a diagnostic for indicating the discrepancy between the proposal distribution and the target distribution. $\hat{k} < 0.7$ indicates that the normalizing flow approximation is close to the target distribution. Values of $\hat{k}$ above the 0.7 threshold are indicative of potential issues and reported in red. As shown in Table A.9 and A.10, PSIS-$\hat{k}$ diagnostics is below the threshold 0.7 for all problems except the multivariate Rosenbrock-Gaussian posterior. However, as we see from the metrics $\Delta$LML, MMTV, and GsKL in Table 1 and Figure A.2($d$), the normalizing flow approximation matches the ground truth target posterior well. We hypothesize that the alarm raised by PSIS-$\hat{k}$ is due to the long tail in multivariate Rosenbrock-Gaussian distribution and PSIS-$\hat{k}$ is sensitive to tail underestimation in the normalizing flow approximation.

In the case of noisy likelihood evaluations or additional likelihood evaluations not available, PSIS cannot be applied. Instead, we can use corner plots (Foreman-Mackey, 2016) to detect algorithm failures. Corner plots visualize posterior samples using pairwise two-dimensional marginal density contours, along with one-dimensional histograms for marginal distributions. For diagnostics purposes, we could overlay training points $\mathbf{X}$ for NFR onto the corner plots to check whether high-probability regions are adequately supported by training data (Li et al., 2024). A common failure mode, known as *hallucination* (De Souza et al., 2022; Li et al., 2024), occurs when flow-generated samples lie far from the training points, indicating that the flow predictions cannot be trusted. Figure A.1 provides an example of such a diagnostic plot. The failure case shown in Figure A.1($a$) was obtained by omitting the flow priors as done in the ablation study (Appendix A.7).

Table A.10: PSIS diagnostics. Both the median and the 95% confidence interval (CI) of the median are provided. We show the PSIS-$\hat{k}$ statistic computed with 100, 1000, and 2000 proposal samples. $\hat{k} > 0.7$ indicates potential issues and is reported in red. (BADS)

| Problem | PSIS-$\hat{k}$ (100) | PSIS-$\hat{k}$ (1000) | PSIS-$\hat{k}$ (2000) |
|---|---|---|---|
| Multivariate Rosenbrock-Gaussian | 0.54 [0.33,0.79] | 0.61 [0.46,0.97] | 0.77 [0.44,1.1] |
| Lumpy | 0.41 [0.23,0.46] | 0.37 [0.32,0.45] | 0.32 [0.25,0.45] |
| Lotka-Volterra model | 0.41 [0.22,0.58] | 0.35 [0.27,0.38] | 0.35 [0.26,0.45] |
| Multisensory | 0.52 [0.20,0.58] | 0.34 [0.27,0.41] | 0.36 [0.26,0.47] |

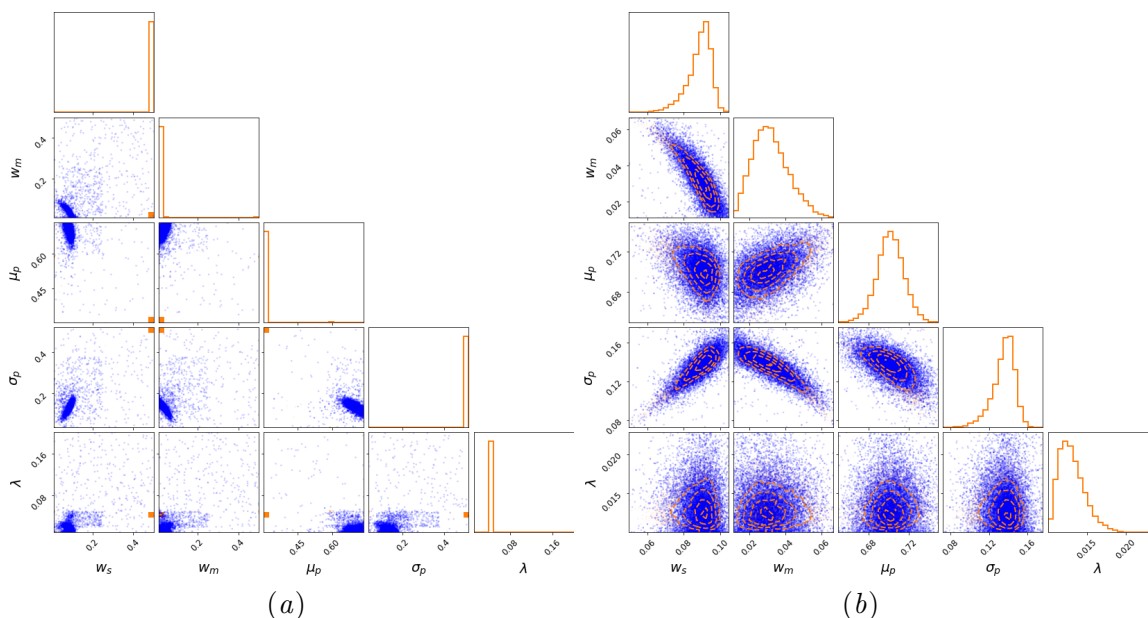

(a)         (b)

Figure A.1: Diagnostics using corner plots. The orange density contours represent the flow posterior samples, while the blue points indicate training data for flow regression. (a) The flow's probability mass escapes into regions with few or no training points, highlighting an unreliable flow approximation. (b) The high-probability region of the flow is well supported by training points, indicating that the qualitative diagnostic check is passed.

## A.9. Visualization of posteriors

As an illustration of our results, we use *corner* plots (Foreman-Mackey, 2016) to visualize posterior samples with pairwise two-dimensional marginal density contours, as well as the 1D marginals histograms. In the following pages, we report example solutions obtained from a run for each problem and algorithm (Laplace, BBVI, VSBQ, NFR).[9] The ground-truth posterior samples are in black and the approximate posterior samples from the algorithm are in orange (see Figure A.2, A.3, A.4, A.5, and A.6).

---

9. Both VSBQ and NFR use the log-density evaluations from CMA-ES, as described in the main text.

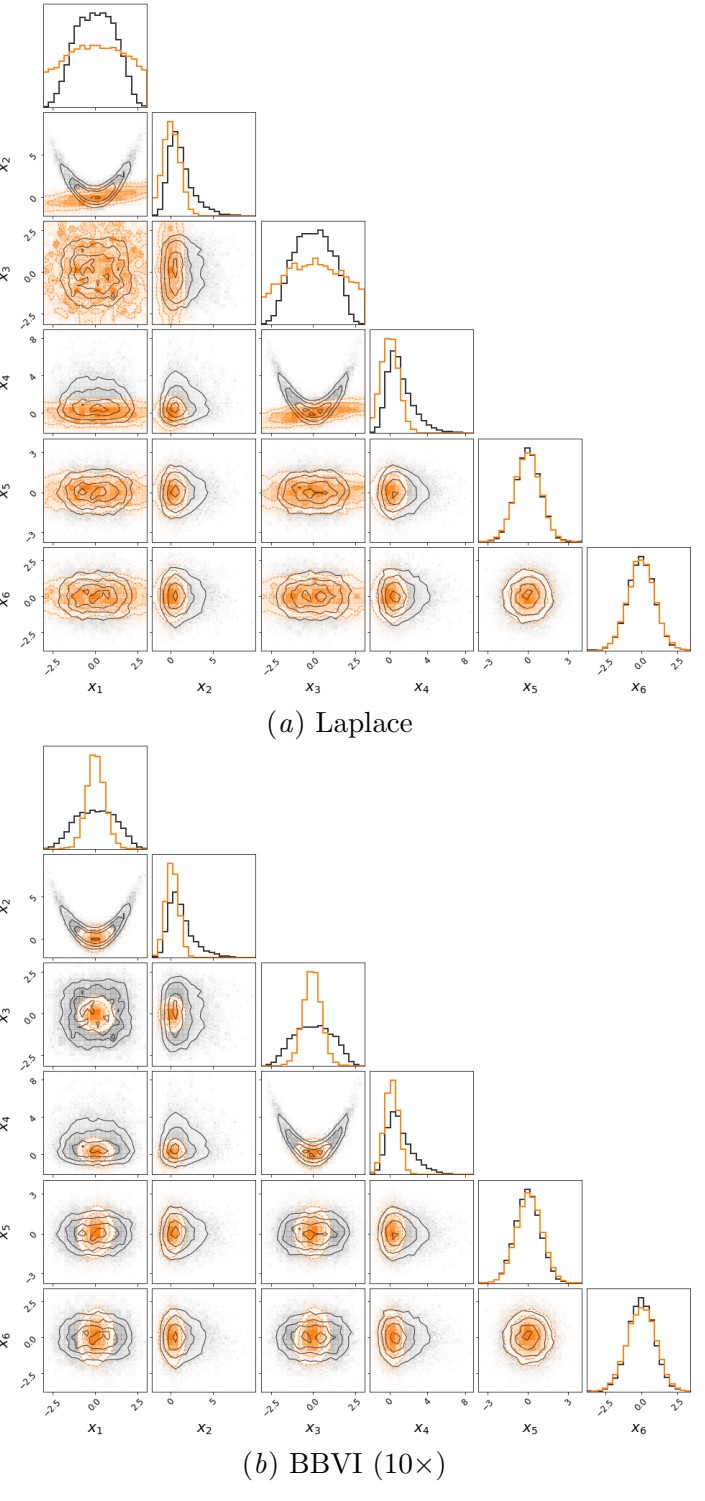

($a$) Laplace

($b$) BBVI (10×)

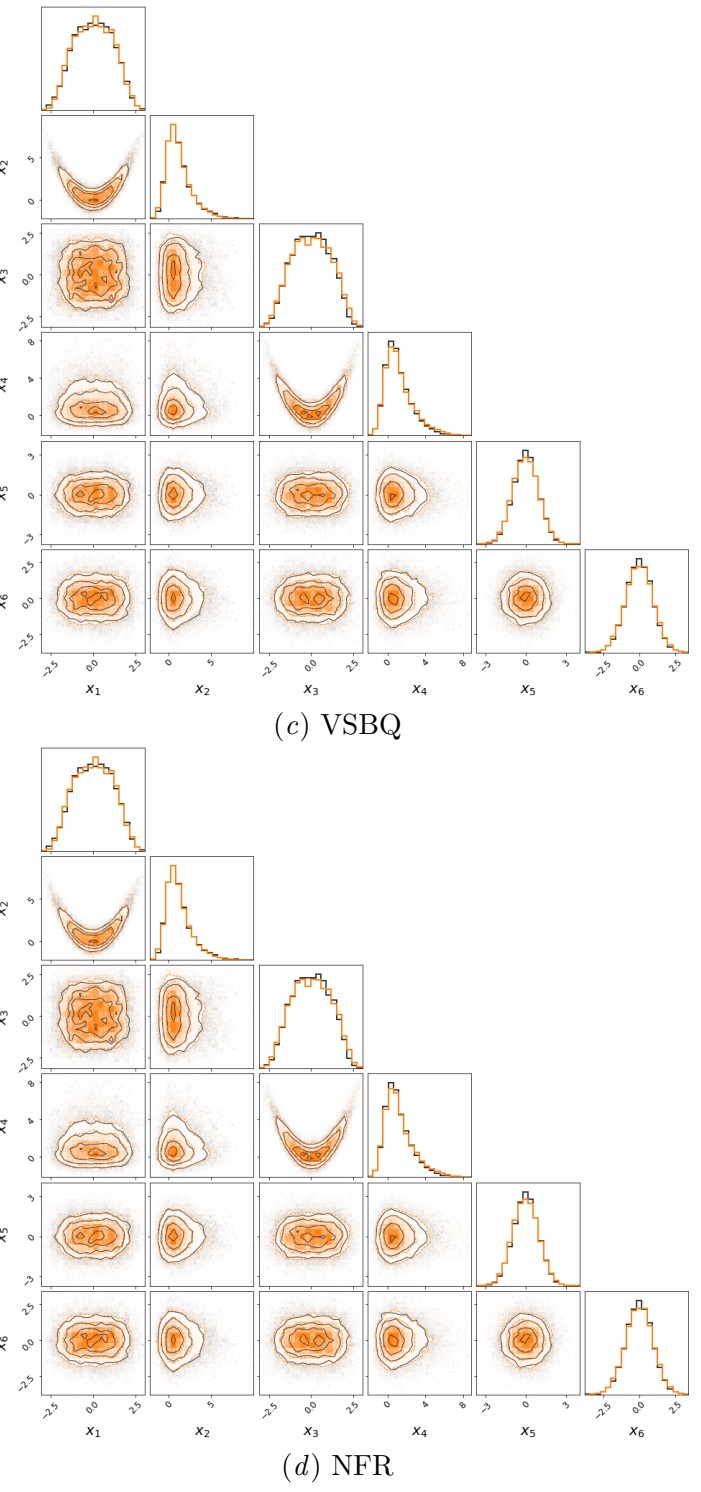

(c) VSBQ

(d) NFR

Figure A.2: Multivariate Rosenbrock-Gaussian ($D = 6$) posterior visualization. The orange density contours and points in the sub-figures represent the posterior samples from different algorithms, while the black contours and points denote ground truth samples.

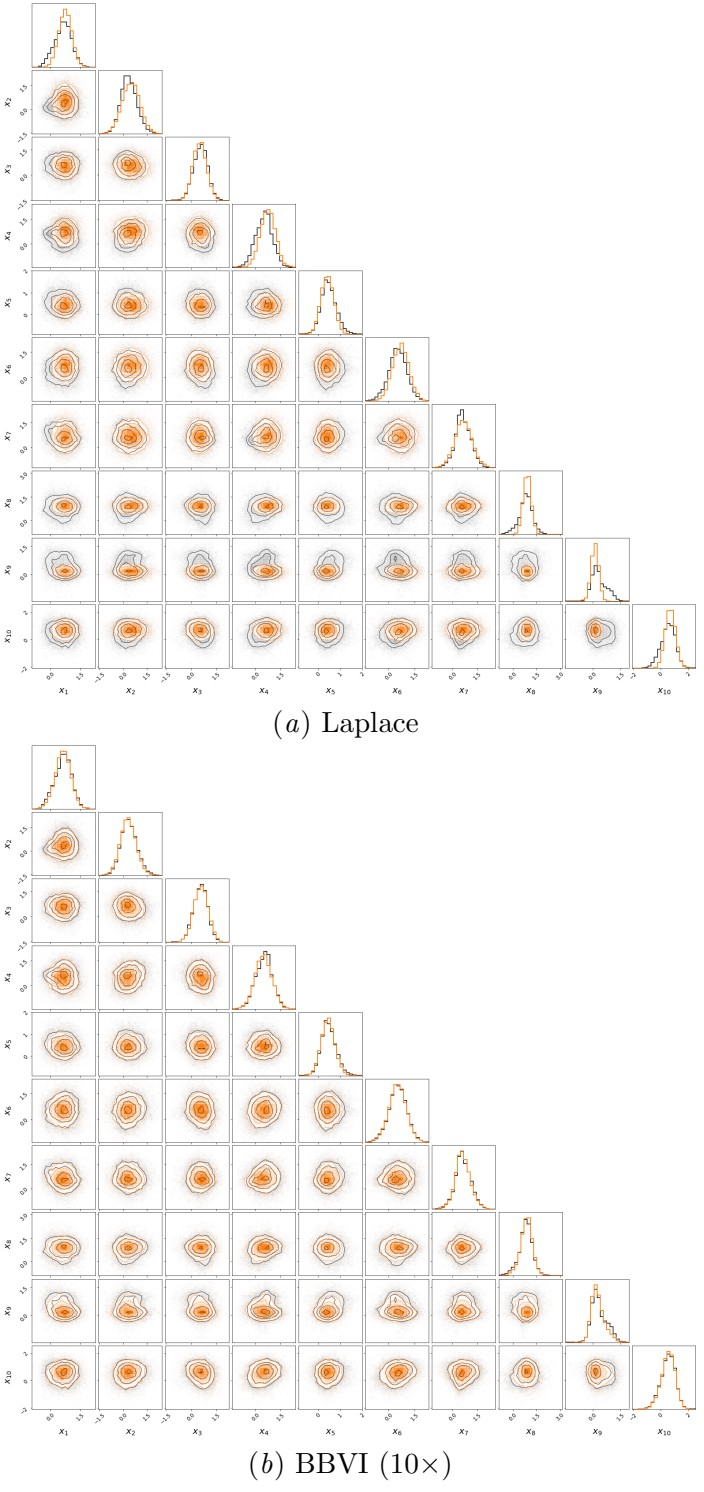

(*a*) Laplace

(*b*) BBVI (10×)

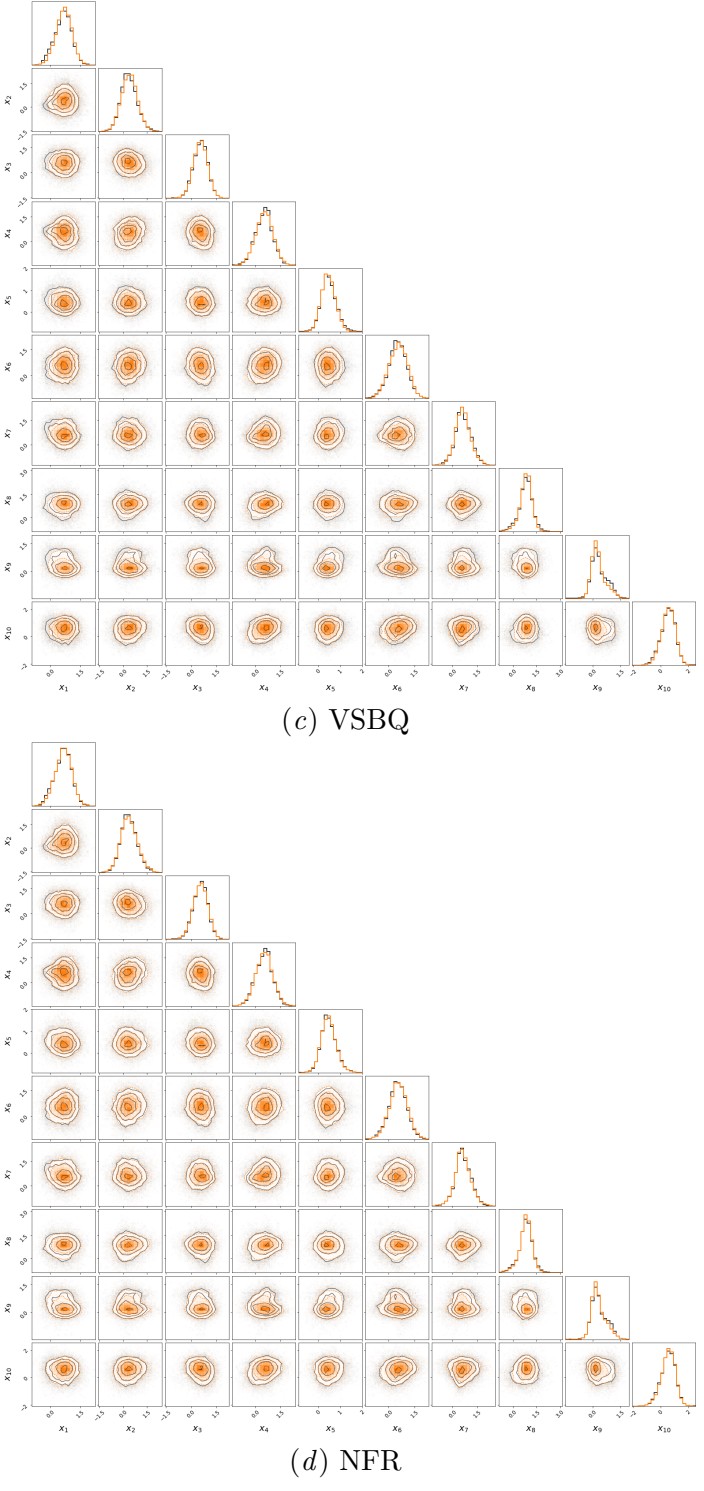

(*c*) VSBQ

(*d*) NFR

Figure A.3: *Lumpy* ($D = 10$) posterior visualization. The orange density contours and points in the sub-figures represent the posterior samples from different algorithms, while the black contours and points denote ground truth samples.

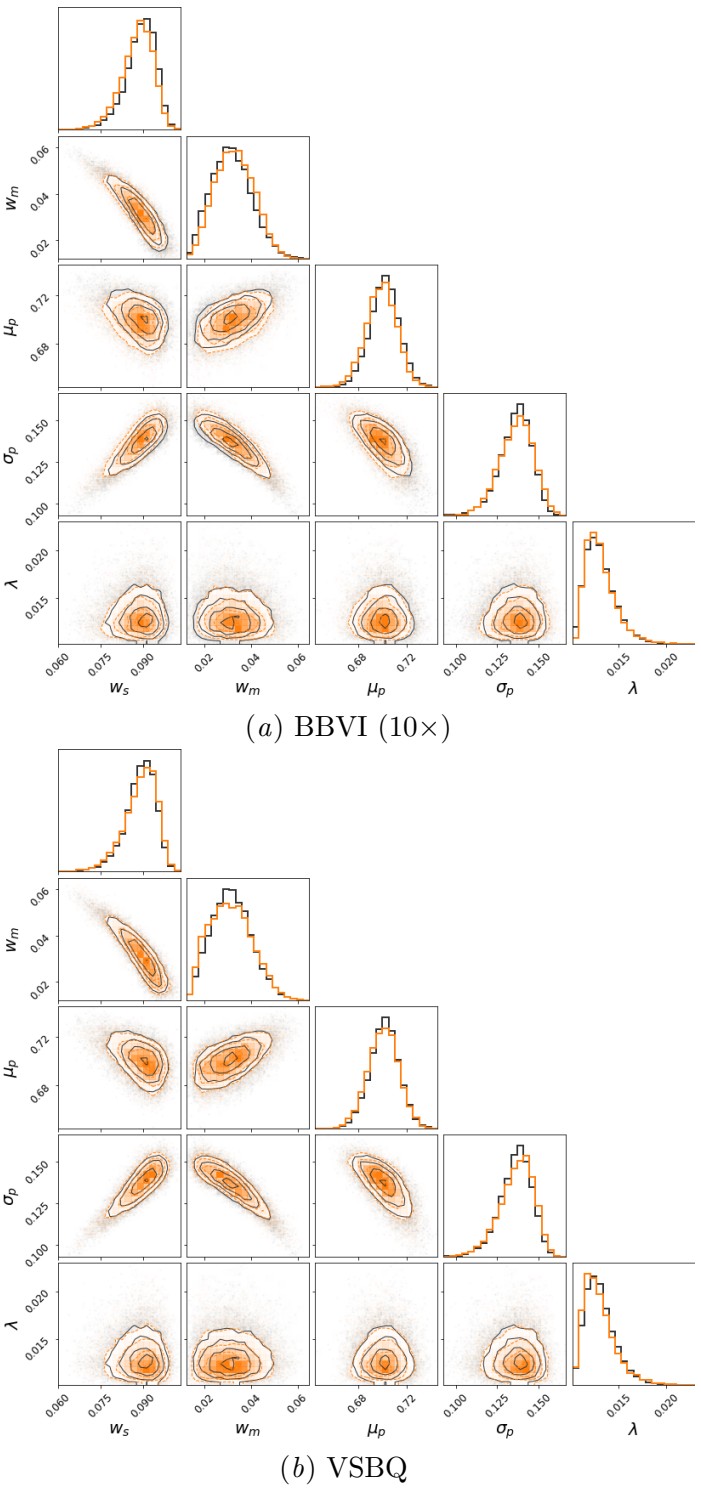

(a) BBVI (10×)

(b) VSBQ

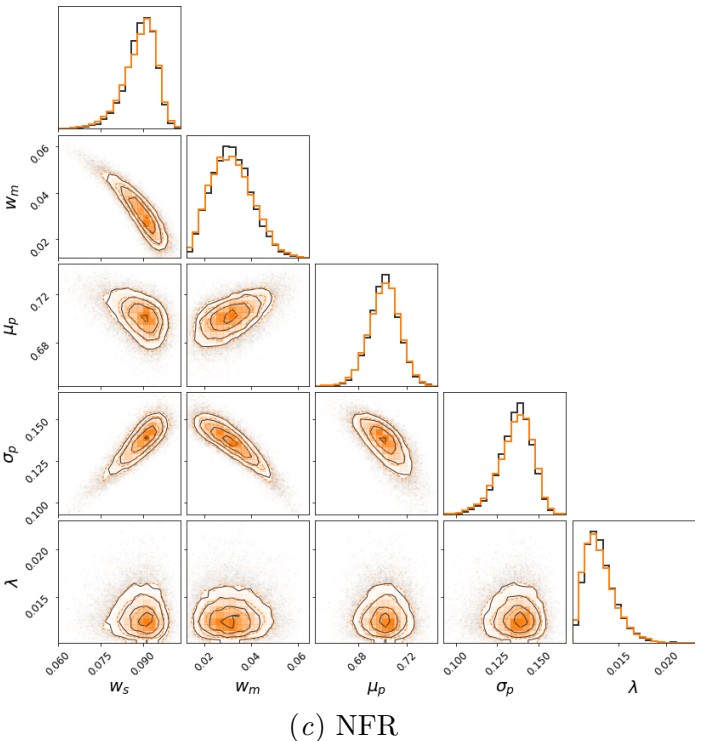

(*c*) NFR

Figure A.4: Bayesian timing model ($D = 5$) posterior visualization. The orange density contours and points in the sub-figures represent the posterior samples from different algorithms, while the black contours and points denote ground truth samples.

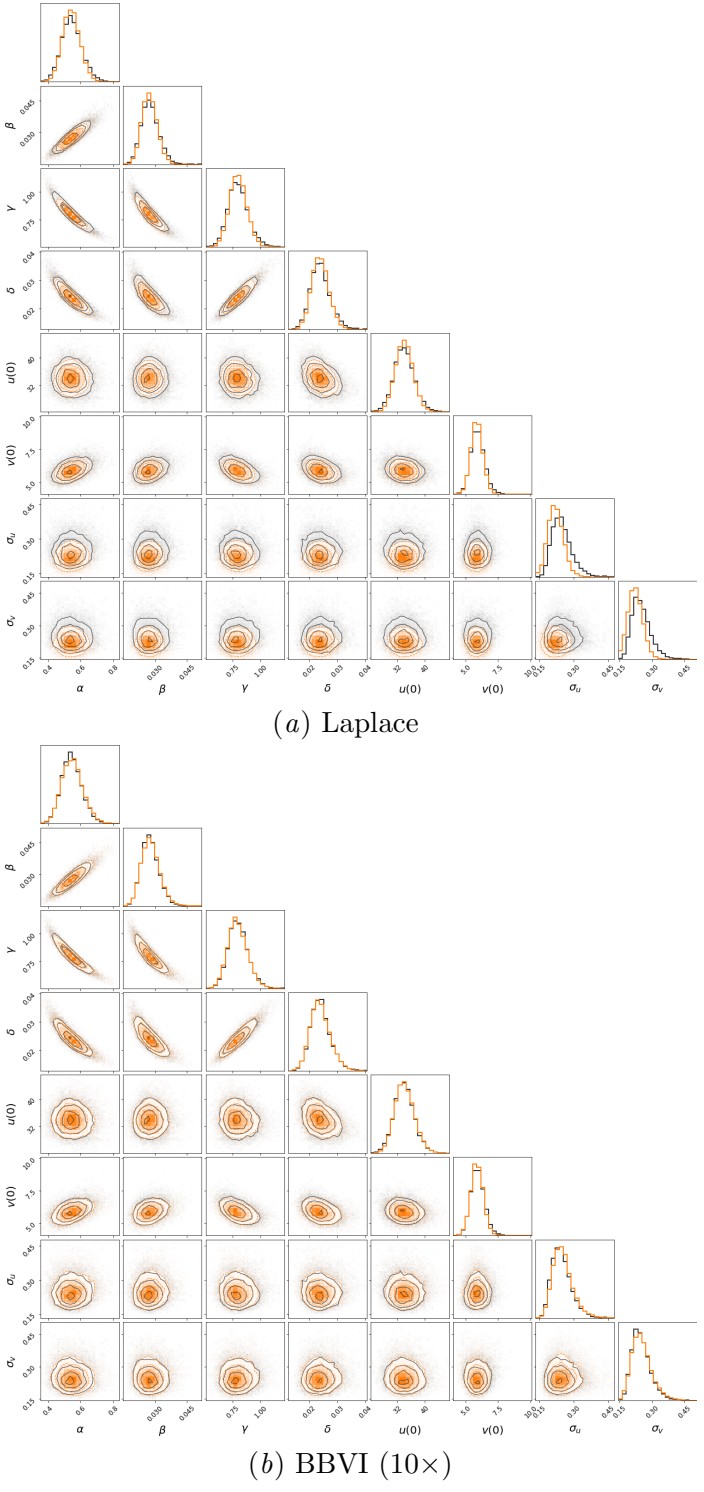

(*a*) Laplace

(*b*) BBVI (10×)

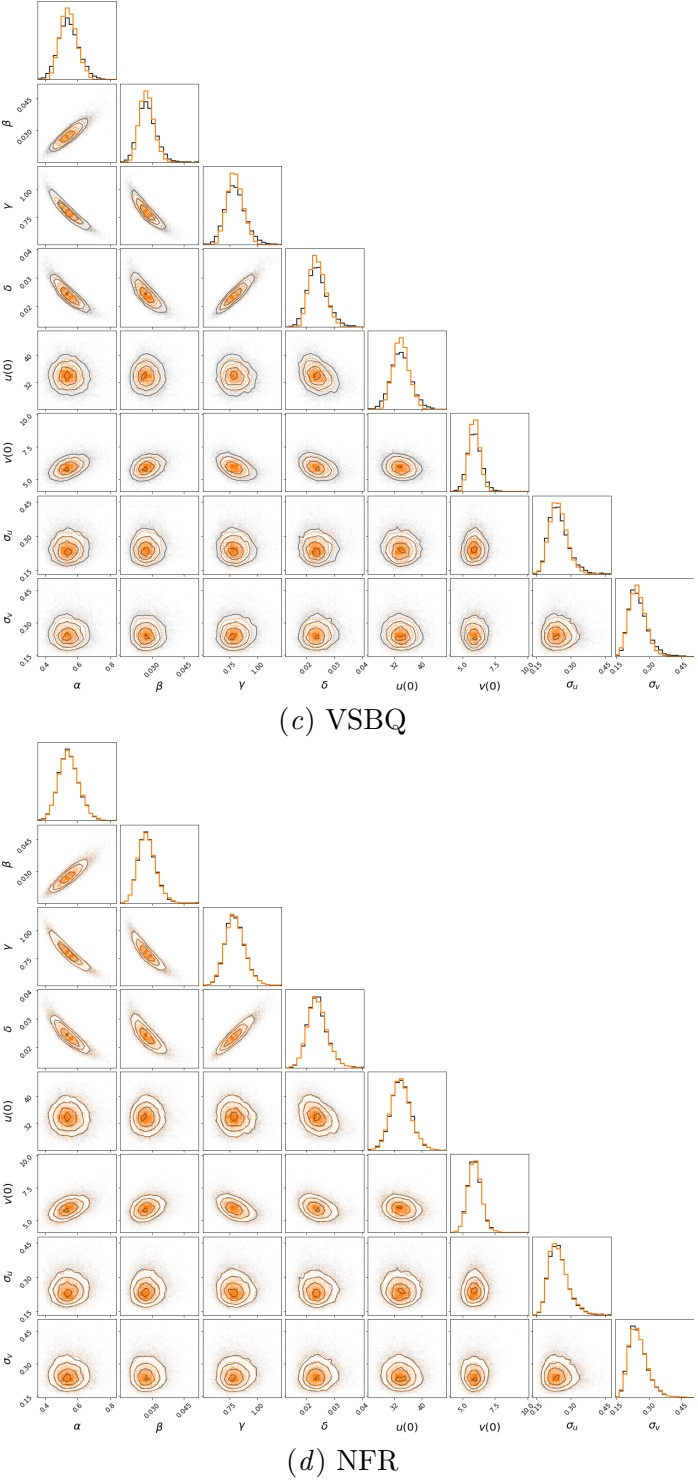

($c$) VSBQ

($d$) NFR

Figure A.5: Lotka-Volterra mode ($D = 8$) posterior visualization. The orange density contours and points in the sub-figures represent the posterior samples from different algorithms, while the black contours and points denote ground truth samples.

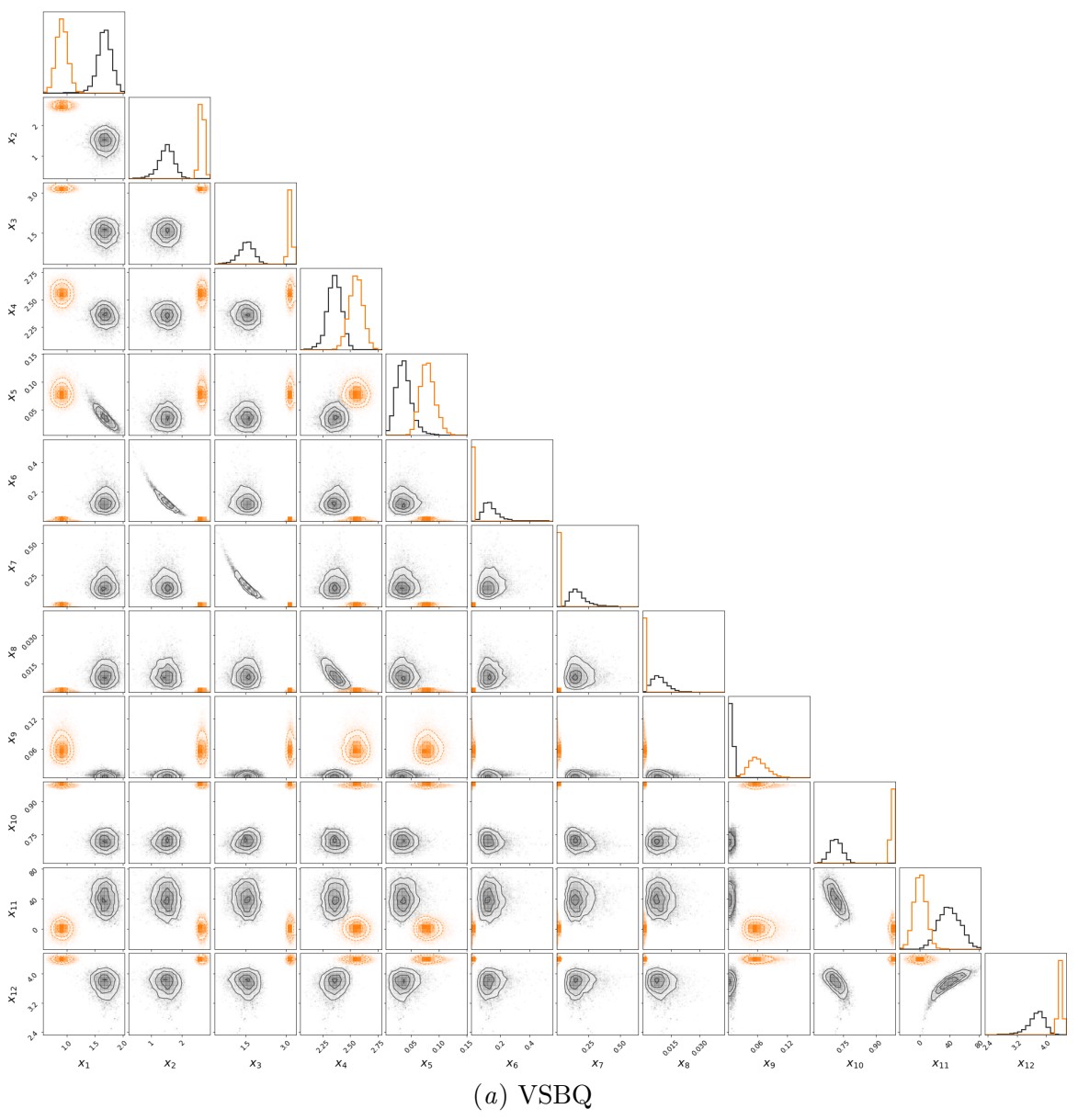

(*a*) VSBQ

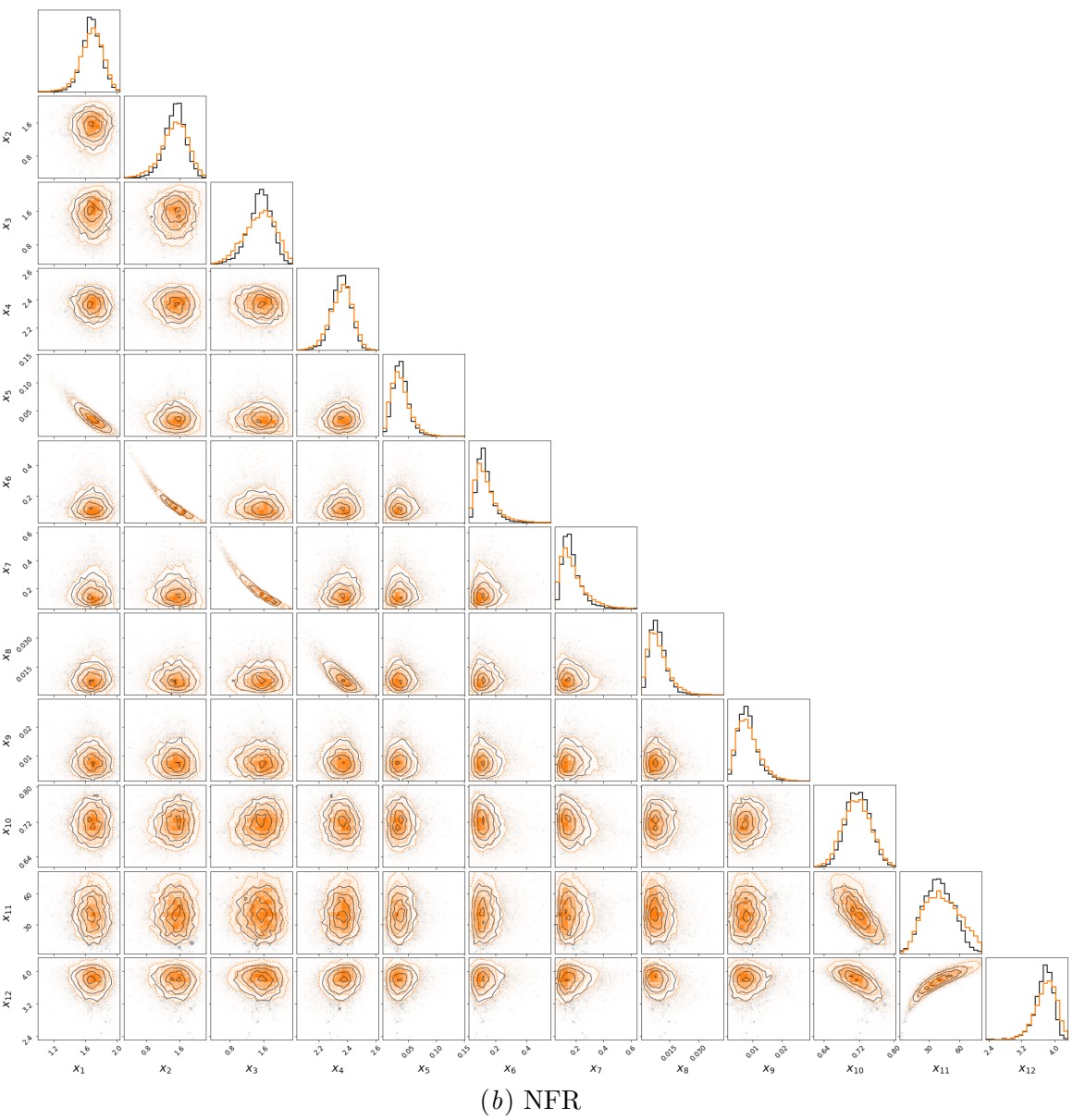

(b) NFR

Figure A.6: Multisensory ($D = 12$) posterior visualization. The orange density contours and points in the sub-figures represent the posterior samples from different algorithms, while the black contours and points denote ground truth samples.

