# OpenReview forum: "Normalizing Flow Regression for Bayesian Inference with Offline Likelihood Evaluations"
_approximateinference.org/AABI/2025/Proceedings_Track — AABI 2025 Proceedings Track_

### Official Review · Reviewer_K3kE · 2025-02-25
**Review of "Normalizing Flow Regression for Bayesian Inference with Offline Likelihood Evaluations"**

**Rating:** 6
**Confidence:** 2

**Review:**

In this paper, the authors introduce normalizing flow regression for approximate posterior inference. A key advantage of the new proposal is that it works directly with existing samples of unnormalized log likelihood evaluation without additional sampling steps. The method is clearly described and evaluated using simulation and real world data. The paper also includes a detailed appendix with useful details and recommendations to practitioners.

Pros:
+ The method is clearly documented and could be the basis of future approximate Bayesian methods using normalizing flows.
+ The experiments are detailed and convincingly demonstrate the effectiveness of the method

Cons:
+ The practical usage of the model seems quite limited right now given it only works with models with dimension < 20. In such low-dimensional problems, I would expect traditional methods such as MCMC can usually be implemented efficiently under comprable amounts of likelihood evaluation (3000 * D). Can the authors provide more convincing examples where simpler alternative does not work?

+ The target density of the problems considered in the numerical examples are largely close to Gaussian (and usually uncorrelated Gaussian), based on the plots in A9. The banana shaped density in Figure 4 is more interesting. There are many widely studied problems where naive MCMC can fail (e.g., multi-modal, donut-shaped, funnels, ...) The results will be more convincing if the authors can demonstrate their approach to those more challenging problems.

+ Related to the above, the suitability of priors and likelihood function under the challenging densities above is less clear.

---

### Official Review · Reviewer_EiSZ · 2025-02-28

**Rating:** 7
**Confidence:** 4

**Review:**

The paper presents an offline Bayesian inference algorithm using Normalizing Flow Regression (NFR), which enables efficient offline Bayesian inference by approximating posteriors from existing likelihood evaluations. It employs regression-based normalizing flows, tailored priors, and annealed optimization for improved performance. NFR outperforms or matches existing methods while reducing computational costs, offering a practical alternative to MCMC and Variational Inference.
NFR leverages normalizing flows for flexible and invertible probability transformations, optimizes log-density estimates by integrating prior knowledge and uncertainty-aware likelihoods, and employs an annealing strategy to gradually refine posterior approximations, ensuring stability and accuracy.
NFR outperforms alternatives in posterior approximation on synthetic datasets like Rosenbrock-Gaussian and Lumpy distributions. It accurately reconstructs posteriors in neuroscience and biology applications with limited likelihood evaluations. Compared to BBVI and VSBQ, NFR is more robust and requires fewer evaluations for accurate Bayesian inference.

---

### Official Review · Reviewer_sVDc · 2025-02-28
**A modern technique for the intractable likelihood problem**

**Rating:** 7
**Confidence:** 2

**Review:**

The paper applies normalizing flow regression for approximating posterior distributions without extensive iterations or additional sampling. An interesting aspect of this approach is to treat also the normalizing constant as a parameter with a prior.

One major concern about this method regards the question of whether it is usually easy to obtain a meaningful set of training points (likelihood evaluations with variances). Then, the "offline" nature of the method can be questioned as a disadvantage because it does not allow adaptively choosing 'smart' points for x to learn better the likelihood function.

The paper is rich in numerical examples. The presentation of the model and the method is carried out with sufficient rigor and clarity. The Appendix offers some follow-up discussions and experiments such as an ablation study. It would be nice to mention the limitations of the paper in its main body.

---

### Official Review · Reviewer_E6V3 · 2025-02-28
**Normalizing flows for un-normalized density estimation**

**Rating:** 7
**Confidence:** 4

**Review:**

This paper is using normalizing flows to provide normalized estimates of a target density known only up to a multiplicative constant and at a given set of points.  Their motivating use case is for surrogate posteriors, noting that other standard methods are either formally restrictive (e.g. the Laplace approximations), produce unnormalized approximations (e.g. Gaussian processes), or require actively querying the unnormalized target density (e.g. BBVI).  However, I think a good solution to this general problem could apply more generally, e.g., for producing density and normalizing constant estimates from MCMC for sensitivity analysis or model selection.

All in all, this is an interesting and important problem, this approach is novel to the best of my knowledge, and the experiments are promising.  I have some doubts about the number of arbitrary and difficult-to-reason-about features of the procedure, but these concerns don't rise to the level of rejecting the paper.

Their approach has several distinctive features relative to standard normalizing flow problems:
(A) The data is not a set of draws from the target distribution, but a (possibly highly idiosyncratic) set of evaluations, such as the sequence of points along an optimization path, and so
(B) The NF and target will differ, and they need a principled way to penalized deviations.
(C) They are trying to match the NF density to the target density, but the target density is known only up to a constant, which must be estimated,

They deal with (B) by assuming that the density is observed with some minimal noise, and crafting a somewhat ad-hoc custom loss function that avoids overfitting large, negative log likliehood values that do not matter for inference.  As part of this procedure you need to set hyperparameters like a minimal density to attempt to fit, which seem difficult to think about carefully in higher dimensions.  This whole construct, though apparently necessary, seems especially artificial if the target density is not measured with any uncertainty.

I'm particularly concerned about (C), since this seems like a fundamental problem that the present work does not take on in a very transparent way.  They correctly observe that there is a non-identifiability problem, but this problem seems deeper to me than is acknolwedged in the paper --- the value of the inferred normalizing constant essentially determines the smoothness of the fit.  Consider fitting unnormalized values (x_n, y_n) which are { (1, 5), (2, 6) } versus { (1, 105), (2, 106) }.  In principle these two datasets should be indistinguishable, since the y_n are supposed to be the target density only up to a known constant.  However, there is truly no information about whether the target density is smooth or has two spikes, and which you choose will depend on the prior.

When you fit a surrogate with a GP, the smoothness parameter is clear --- it's typically part of the kernel function.  How is the smoothness parameter determined in the present case?  The authors put priors on the neural network parameters in an effort to force the fit to be close to the proposal distribution, and it is notoriously difficult to think about the implications of NN parameter priors.  Figure 3 seems represent the authors' attempt to calibrate these priors, and it seems extremely ad-hoc for such an essential part of this procedure.  Since this is such a conceptually central point, I would have liked more careful thought and attention given to the prior specification.

I would also have liked to see computational costs for the different procedures in the experiments section.

All told, I think this is a nice contribution, and I recommend acceptance.

---

### Official Review · Reviewer_25Bh · 2025-02-28
**A regression method for offline posterior approximations.**

**Rating:** 7
**Confidence:** 3

**Review:**

This paper introduces normalizing flow regression for posterior approximation in Bayesian inference. This work specifically targets the case of offline inference, where a number of likelihood/density evaluations have already been performed, e.g. when calculating an MLE, and uses these evaluations to estimate the entire posterior.

The basis of the method is normalizing flow, which defines an invertible transformation between finite dimensional spaces of dimension D. They specifically use the masked auto regression flow, which decomposes the transformation into scale and shift components, the arguments to which have been parameterized by neural networks. The regression model is trained via MAP estimation. The likelihood function is assumed to be a Tobit likelihood, which has the effect of censoring the likelihood below some threshold. The posterior fitting is performed iteratively via annealed optimization.

A number of experiments are then performed comparing the proposed method to Laplace approximation, black-box variational inference (BBVI), and variational sparse Bayesian quadrature (VSBQ) using a series of simulated and real-world data sets. The proposed method generally performs as well or better than the 3 other methods based on the same training data.

The manuscript is clearly written and the proposed method is thoroughly described. The benchmarks are extensive and thoroughly described, and in general provide solid evidence the method performs well over a range of different problems. Like any offline method, as the authors acknowledge "the training dataset needs to sufficiently cover regions of non-negligible probability mass". One question I would have is whether in the case of missing regions of non-negligible probability mass could the regression model be used to informatively select new points for likelihood/posterior approximation? This is beyond the scope of the current work but could potentially expand the application beyond the offline setting.

The other major question I have regards the many choices made vis-a-vis the posterior surrogate, e.g. Tobit likelihood, the flow transformation priors, the use of annealed optimization, etc. The ablation study reasonably demonstrates these choices outperform some obvious alternatives, but it does make this reviewer wonder if there are perhaps similar potential estimation errors inherent in these choices that are simply not apparent in the current set of test problems. More understanding of how to characterize situations in which some level of performance could be guaranteed would be very useful, albeit clearly a significant project in its own right.

Post-rebuttal
After reading the authors responses and other reviews I'm electing to keep my score the same. The scores were quite consistent across reviewers, four of 7 and one of 6, so it seems we are largely in agreement on the merits of the manuscript.

---

### Meta-Review · Area_Chair_Ksf2 · 2025-03-16

**Recommendation:** Accept
**Confidence:** 4

**Metareview:**

This paper proposes to use a normalizing flow (NF) as a regression model. Unlike standard neural networks (NNs), it approximates posterior beliefs; unlike weight-space Bayesian NNs, it is much flexible in approximating the posterior. The authors discussed the challenges in using NF for regression, e.g. likelihood and prior specification, optimization, etc.

All reviewers agree that this is good work and should be accepted.

---

### Decision · Program_Chairs · 2025-03-18

Accept